



# Chemical ionization of clusters formed from sulfuric acid and dimethylamine or diamines

Coty N. Jen[1,2*], Jun Zhao[1,3], Peter H. McMurry[1], David R. Hanson[4]

[1]Department of Mechanical Engineering, University of Minnesota – Twin Cities, 111 Church St. SE, Minneapolis, MN, 55455, USA
[2] now at Department of Environmental Science, Policy, and Management, University of California, Berkeley, Hilgard Hall, Berkeley, CA, 94720
[3] now at Institute of Earth Climate and Environment System, Sun Yat-sen University, 135 West Xingang Road, Guangzhou 510275, China
[4]Department of Chemistry, Augsburg College, 2211 Riverside Ave., Minneapolis, MN, 55454, USA

*Correspondence to: Coty N. Jen (jenco@berkeley.edu)

*Abstract:* Chemical ionization (CI) mass spectrometers are used to study atmospheric nucleation by detecting clusters produced by reactions of sulfuric acid and various basic gases. These instruments typically use nitrate to deprotonate and thus chemically ionize the clusters. In this study, we compare cluster concentrations measured using either nitrate or acetate. Clusters were formed in a flow reactor from vapors of sulfuric acid and dimethylamine, ethylene diamine, tetramethylethylene diamine, or butanediamine (also known as putrescine). These comparisons show that nitrate is unable to chemically ionize clusters with high base content. In addition, we vary the ion-molecule reaction time to probe ion processes which include proton-transfer, ion-molecule clustering, and decomposition of ions. Ion decomposition upon deprotonation by acetate/nitrate was observed. More studies are needed to quantify to what extent ion decomposition affects observed cluster content and concentrations, especially those chemically ionized with acetate since it deprotonates more types of clusters than nitrate.

Model calculations of the neutral and ion cluster formation pathways are also presented to better identify the cluster types that are not efficiently deprotonated by nitrate. Comparison of model and measured clusters indicate that sulfuric acid dimer with two diamines and sulfuric acid trimer with two or more base molecules are not efficiently chemical ionized by nitrate. We conclude that acetate CI provides better information on cluster abundancies and their base content than nitrate CI.

**Introduction:**

Atmospheric nucleation is an important source of global atmospheric particles (IPCC, 2014). In the atmospheric boundary layer, sulfuric acid often participates in nucleation(Weber et al., 1996;Kuang et al., 2008;Kulmala et al., 2004;Riipinen et al., 2007) and its clusters react with other trace compounds to produce stable, electrically neutral molecular clusters; these compounds include ammonia (Kirkby et al., 2011;Coffman and Hegg, 1995;Ball et al., 1999), amines (Almeida et al., 2013;Zhao et al., 2011;Glasoe et al., 2015), water (Leopold, 2011), and oxidized organics (Schobesberger et al., 2013). The primary instruments used for detecting freshly nucleated, sulfuric acid-containing clusters are atmospheric pressure chemical ionization mass spectrometers (CIMS) such as the Cluster CIMS (Zhao et al., 2010;Chen et al., 2012) and the CI atmospheric pressure interface-time of flight mass spectrometer (CI-APi-ToF) (Jokinen et al., 2012). Both mass spectrometers use nitrate to chemically ionize neutral sulfuric acid clusters. Depending upon conditions, $NO_3^-$ core ions generally have one or more $HNO_3$ and possibly several $H_2O$ ligands The signal ratio of the ion cluster to the reagent ion translates to the neutral cluster concentration (Berresheim et al., 2000;Hanson and Eisele, 2002;Eisele and Hanson, 2000).

The amounts and types of ions detected by the mass spectrometer are affected by four key processes: the abundance of neutral clusters, their ability to be chemically ionized, product ion decomposition, and clustering reactions of the product ions (ion-induced clustering, IIC). The first process, neutral cluster formation, follows a sequence of acid-base reactions (Chen et al., 2012;Jen et al., 2014;Almeida et al., 2013;McGrath et al., 2012) whereby sulfuric acid vapor and its clusters react with basic molecules to produce clusters that are more stable than aqueous





sulfuric acid clusters. The concentration of a specific cluster type depends on its stability (i.e. evaporation rates of the
neutral cluster) and the concentrations of precursor vapors (i.e. the formation rate).
Neutral clusters then need to be ionized to be detected with a mass spectrometer. In most prior work, this has
been accomplished by chemical ionization with the nitrate ion whereby the neutral clusters are exposed to nitrate for
a set amount of time known as the chemical ionization reaction time (or ion-molecule reaction time). Chemical
ionization (CI) can be conceptualized as another acid-base reaction where an acid (sulfuric acid) donates a proton to
the basic reagent ion (nitrate, the conjugate base of nitric acid). To illustrate, the CI reaction of an aminated sulfuric
acid dimer, $(H_2SO_4)_2 \cdot DMA$, is shown in Reaction 1.

$$\left(H_2SO_4\right)_2 \bullet DMA \bullet \left(H_2O\right)_x + HNO_3 \bullet NO_3^- \xrightarrow{k_2} HSO_4^- \bullet H_2SO_4 \bullet DMA + 2HNO_3 + x\left(H_2O\right) \qquad \textbf{Reaction 1}$$

This dimer of sulfuric acid contains a dimethylamine (DMA) molecule and $x$ water molecules. At room temperature,
water molecules evaporate upon ionization or entering the vacuum region and are assumed to not significantly affect
chemical ionization rates. The forward rate constant, $k_2$, is assumed to be the collisional rate coefficient of $1.9 \times 10^{-9}$
$cm^3$ $s^{-1}$ (Su and Bowers, 1973), while the reverse rate constant is zero.
Reaction 1 can be extended to CI reactions for larger neutral clusters of sulfuric acid, with the assumption
that every collision between nitrate and a sulfuric acid cluster results in an ionized cluster. However, Hanson and
Eisele (2002) presented evidence that some clusters of sulfuric acid and ammonia were not amenable to ionization by
$(HNO_3)_{1-2} \cdot NO_3^-$. In addition, Jen et al. (2015) showed that CI with $(HNO_3)_{1-2} \cdot NO_3^-$ leads to significantly lower neutral
concentrations of clusters with 3 or more sulfuric acid molecules and varying numbers of DMA molecules compared
to results using acetate reagent ions. Furthermore, neutral cluster concentrations detected using acetate CI are in overall
better agreement with values measured using a diethylene glycol mobility particle sizer (DEG MPS). As no other
experimental conditions changed except the CI reagent ion, we hypothesized that nitrate's lower proton affinity than
acetate renders it less able to chemically ionize clusters that contain nearly equal amounts of sulfuric acid and base.
Poor CI efficiency reduces the amount and types of ions detected by the mass spectrometer.
After neutral clusters are ionized, the resulting ion may decompose. Experimental studies have shown ion
decomposition in the ammonia-sulfuric acid system at 275 K (Hanson and Eisele, 2002), and computational chemistry
studies present evaporation rates of ion clusters of sulfuric acid with various bases on the order of the CI reaction time
used here (Kurtén et al., 2011;Lovejoy and Curtius, 2001;Ortega et al., 2014). For example, these studies predict an
evaporation rate, $E_d$ (Reaction 2), of DMA from a sulfuric acid dimer ion with 1 DMA molecule of ~100 $s^{-1}$ at 298 K
(Ortega et al., 2014).

$$HSO_4^- \bullet H_2SO_4 \bullet DMA \xrightarrow{E_d} HSO_4^- \bullet H_2SO_4 + DMA \qquad \textbf{Reaction 2}$$

Experimental observations at room temperatures have never seen the aminated sulfuric acid dimer ion, even at CI
reaction times as short as a few ms. Thus, the decomposition rate is likely even faster than the computed value of ~100
$s^{-1}$ at 298 K (Ortega et al., 2014).
Ion clusters can also be produced by ion-induced clustering (IIC) whereby the bisulfate ion ($HSO_4^-$), formed
by CI of sulfuric acid monomer, further reacts with $H_2SO_4$ (with ligands) and larger clusters. Charged clusters can also
cluster with neutrals to form larger ion clusters. The signal due to these IIC products must be subtracted from the
observed signals to determine neutral cluster concentrations. Specifically, the sulfuric dimer ion can be formed via the
IIC pathway given in Reaction 3, with ligands not shown.

$$HSO_4^- + H_2SO_4 \xrightarrow{k_{21}} HSO_4^- \bullet H_2SO_4 \qquad \textbf{Reaction 3}$$

The forward rate constant, $k_{21}$, is the collisional rate constant of $2 \times 10^{-9}$ $cm^3$ $s^{-1}$ because this reaction involves switching
ligands between the two clusters. Both reactants also contain water, nitrate, and/or base ligands that detach during
measurement. IIC-produced dimer signal interferes with the CI detected neutral dimer but can be calculated from
measured sulfuric acid vapor concentrations and CI reaction times (Chen et al., 2012;Hanson and Eisele, 2002).





IIC can also produce larger clusters, but in general its contribution is less than for the dimer, even if all rates
are assumed to be collisional. Furthermore, bisulfate may not efficiently cluster with chemically neutralized sulfate
salt clusters formed by reactions of sulfuric acid and basic compounds. If so, assuming the collisional rate constant
for all IIC-type reactions would lead to an over-correction of the neutral cluster concentrations.
Measured CIMS signals reflect the combined influences of all these processes, with each occurring on time
scales that depend on the chemistry, experimental parameters, and techniques. Assuming a process is either dominant
or negligible can lead to large errors in reported neutral cluster compositions and concentrations. Here, neutral cluster
formation, chemical ionization, IIC, and ion decomposition are examined experimentally and theoretically to
determine the influence of each process on the abundance of ion clusters composed of sulfuric acid and various bases.
These bases include DMA, ethylene diamine (EDA), trimethylethylene diamine (TMEDA), and butanediamine (also
known as putrescine, Put). The diamines, recently implicated in atmospheric nucleation, react with sulfuric acid vapors
to very effectively produce particles compared to monoamines (Jen et al., 2016). We present observations that 1) show
a clear difference between acetate and nitrate CI for all clusters larger than the sulfuric acid dimer with any of the
bases, 2) provide evidence of ion decomposition, and (3) identify specific bases that influence the detectability of the
dimer neutral clusters. Also presented are modeling results that help elucidate specific processes that influence
measurement: neutral cluster formation pathways, cluster types that do not undergo nitrate CI, and clusters that are
formed by IIC.
**Method:**
Sulfuric acid clusters containing either DMA, EDA, TMEDA, or Put were produced in a flow reactor that
allows for highly repeatable observations (see Jen et al. (2014) and Glasoe et al. (2015)). Glasoe et al. (2015) showed
that the system has a high cleanliness level: 1 ppqv level or below for amines. Each amine was injected into the flow
reactor at a point to yield ~3 s reaction time between the amine and sulfuric acid (see Jen et al. (2014) for a schematic).
The initial sulfuric acid concentration ($[A_1]_o$) before reaction with basic gas was controlled at specified concentrations.
The base concentration, [B], was measured by the Cluster CIMS in positive ion mode (see SI of Jen et al. (2014) for
further details) and confirmed with calculated concentrations (Zollner et al., 2012;Freshour et al., 2014). The dilute
amines were produced by passing clean nitrogen gas over either a permeation tube (for DMA and EDA) or a liquid
reservoir (TMEDA and Put), and further diluted in a process described in Zollner et al. (2012). The temperature of the
flow reactor was held constant throughout an experiment but varied day-to-day from 296-303 K to match room
temperature. This was done to minimize thermal convection which induces swirling near the Cluster CIMS sampling
region. The relative humidity was maintained at ~30%, and measurements were done at ambient pressure (~0.97 atm).
Total reactor $N_2$ flow rate was 4.0 L/min at standard conditions of 273 K and 1 atm.
Two types of experiments were conducted: one set where specific base, base concentration ([B]), and $[A_1]_o$
were varied at constant CI reaction time (similar to those in in Jen et al. (2014)), and the second set where CI reaction
time was varied for a subset of reactant conditions (see Hanson and Eisele (2002) and Zhao et al. (2010)). The resulting
concentrations were measured with the Cluster CIMS using either nitrate or acetate as the CI reagent ion. Nitrate and
acetate were produced either by passing nitric acid or acetic anhydride vapor over Po-210 sources. Separate Po-210
sources and gas lines were used for the acetate and nitrate to avoid cross-contamination. The reagent ions for nitrate
CI was $(HNO_3)_{1-2} \cdot NO_3^-$, and the reagent ions for acetate CI were $H_2O \cdot CH_3CO_2^-$, $CH_3CO_2H \cdot CH_3CO_2^-$, and $CH_3CO_2^-$
(in order of abundance). The inferred neutral cluster concentrations were calculated from the CI reaction time,
measured and extrapolated mass-dependent sensitivity (see Supporting Information), and the assumed collisional rate
constant between CI ion and sulfuric acid clusters (see Jen et al. (2014) and (2015) for a discussion on the data
inversion process). The CI reaction time, $t_{CI}$, was determined from the inlet dimensions and electric field strength
inside the sampling region; for this set of experiments, $t_{CI}$ was fixed at 18 ms for nitrate and 15 ms for acetate.
Varying $t_{CI}$ at fixed [B] and $[A_1]_o$ was achieved by changing the electric field used to draw ions across the
sample flow into the inlet. Similar experiments have been performed with other atmospheric pressure, CI mass





spectrometer inlets (Hanson and Eisele, 2002;Zhao et al., 2010;Chen et al., 2012) with the detailed mathematical
relationship between $t_{CI}$ and ion signal ratios developed more in depth in the following sections and the SI.

**Acetate vs. Nitrate Comparison:**

Figure 1 (a and c) compare inferred cluster concentrations derived from measured signals (assuming the
collisional rate constant, $k_c$, and no ion breakup) using acetate (red squares) and nitrate (black triangles) reagent ions
at a constant $[A_1]_o \sim 4 \times 10^9$ cm$^{-3}$ for two different [DMA]. The grouped points represent clusters that contain equivalent
number of sulfuric acid molecules ($N_1$ is the monomer, $N_2$ is the dimer, etc.) but with different number of DMA
molecules (e.g., $A_4 \cdot DMA_{0-3}$ where A is sulfuric acid). The number of base molecules in each cluster is given by the
grouping bracket. Since the tetramers and pentamers have similar mass ranges, $N_4$ clusters are given as half-filled
symbols and $N_5$ clusters as outlined symbols. Note, $N_1$ is detected at different masses between the two reagent ions,
with nitrate at 160 amu=$HSO_4^- \cdot HNO_3$ and acetate at 97 amu=$HSO_4^-$. The total cluster concentrations, $[N_m]$, compared
between the two CI ions are shown in Figure 1 (b and d). The notation used here differs slightly from Jen et al. (2014)
such that $[N_m]$ denotes the total concentration for clusters that contain $m$ sulfuric acids molecules (i.e.,
$[N_m]=[A_m]+[A_m \cdot B_1]+[A_m \cdot B_2]...$) and $A_m \cdot B_j$ represents a specific cluster type with $m$ sulfuric acid molecules and $j$ basic
molecules (B). The measured $[N_1]$ and $[N_2]$ obtained using nitrate and acetate are in good agreement for DMA. In the
set of bases studied in Jen et al. (2014) (ammonia, methylamine, DMA, and trimethylamine), DMA is the strongest
clustering agent, and these results reaffirm the accuracy of previously reported values of $[N_1]$ and $[N_2]$ in Jen (2014)
at high $[A_1]_o$.
Figures 2, 3, and 4 show the acetate and nitrate comparison for EDA, TMEDA, and Put, respectively.
Although nitrate appears to consistently detect less $[N_1]$ than with acetate, the estimated systematic uncertainty on
acetate detected $[N_1]$ is higher than with nitrate due to higher background signals detected by acetate, sensitivity for
the low masses (see SI), and possible influence of diamines on the ion throughput in the mass spectrometer. Other
factors that may influence the detected $[N_1]$ are discussed in the SI. The true acetate $[N_1]$ could be up to a factor of 5
lower. Therefore, for monomer clusters formed from diamines, it is difficult to conclude that acetate and nitrate lead
to significant differences in measured $[N_1]$.
Both acetate and nitrate primarily detect the bare dimer, with $[N_2]$ up to a factor of 5 higher with acetate CI
than nitrate. The systematic uncertainties of acetate $[N_2]$ are about a factor of 2-3 for similar reasons to the uncertainties
for $N_1$. These comparisons seem to suggest that for clusters formed from diamines, nitrate does not detect as many
types of $N_2$ as does acetate; however, the large uncertainty in acetate $[N_2]$ prevents a definitive conclusion as to whether
or not nitrate chemically ionizes all types of dimers. More information is gained from experiments that vary $t_{CI}$ as they
are more sensitive to the various formation pathways. These results are presented in the subsequent sections.
Figures 1 through 4 (b and d) clearly show that more of the larger clusters ($N_3$ and higher) were detected by
acetate CI than nitrate. For all bases, the measured $[N_3]$ by acetate is up to a factor of 10 higher than concentrations
measured by nitrate CI. Nitrate detected small amounts of $N_4$ and no $N_5$, likely due to the ionizable fraction of $[N_4]$
and $[N_5]$ falling below detection limits (<$10^5$ cm$^{-3}$). In addition as [B] increases, the differences between acetate and
nitrate cluster concentrations become more pronounced. This likely occurs because sulfuric acid clusters become more
chemically neutral as [B] increases, thereby decreasing their tendencies to donate protons to nitrate ions. The
differences between acetate and nitrate measured cluster concentrations cannot be explained only by the larger
uncertainties in the acetate measurements. The systematic uncertainties in acetate detected larger clusters is at most a
factor of 2 below reported concentrations. Thus, acetate is more efficient than nitrate at chemically ionizing the larger
cluster population.
The large differences between nitrate and acetate measured $[N_3]$ and $[N_4]$ provide information to better
understand recent atmospheric and chamber measurements. Chen et al. (2012) and Jiang et al. (2011) published $[N_3]$
and $[N_4]$ measured in the atmosphere using a larger version of the Cluster CIMS (Zhao et al., 2010). For both studies,
the measurements were conducted using nitrate CI and only at the clusters' bare masses ($A_3$ and $A_4$). Trimer and




tetramer may have been under-detected, though this is uncertain because the atmosphere contains numerous
compounds that may behave differently than DMA and diamines. If the actual concentrations of trimer and tetramer
were higher than those reported by Jiang et al. (2011), then the fitted evaporation rate of $E_3 = 0.4 \pm 0.3$ s$^{-1}$ from Chen et
al. (2012) is too high and the true value would be closer to 0 s$^{-1}$ (collision-controlled or kinetic limit) that was reported
by Kürten et al. (2014) at 278 K. In addition, Kürten et al. measured $[N_3]$ and $[N_4]$ about a factor of 10 lower than the
collision-controlled limit. They attribute this discrepancy to decreased sensitivity for the larger ions, but it could also
be due to inefficient CI by nitrate.
Comparing our results to the CLOUD experiments, the amount of clusters detected via nitrate CI using the
Cluster CIMS differ from those detected by nitrate using the CI-APi-ToF (Kürten et al., 2014). They observed more
ion clusters that contained nearly equal number of sulfuric acid and DMA molecules (e.g., $A_3 \cdot DMA_2$). Our
experiments suggest that such highly neutralized clusters are not efficiently ionized by our nitrate core ions. We do
not fully understand this difference but longer acid-base reaction times, the amount of ligands on the nitrate core ions,
various inlet designs (e.g., corona discharge vs. our Po-210 or high vs. our low flow rates), temperature (278 K
compared to our 300 K), and ion breakup upon sampling may all play a role.

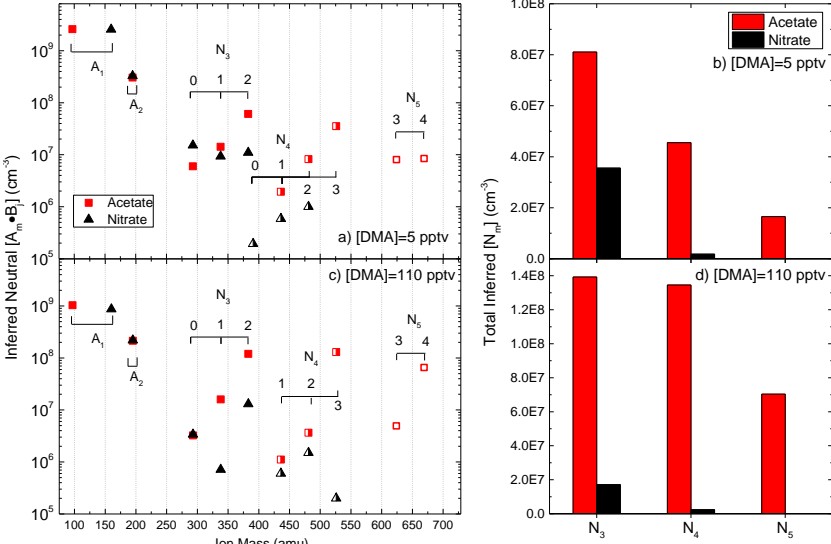

**Figure 1 (a and c) Comparison of specific cluster concentrations ($[A_m \cdot B_j]$) using acetate (red squares) and nitrate (black**
**triangles) reagent ions at two different [DMA] and constant intial sulfuric acid concentration, $[A_1]_0 \sim 4 \times 10^9$ cm$^{-3}$. Each cluster**
**species is shown at its ion mass. The brackets represent the number of DMA molecules in a cluster with a given number of**
**sulfuric acid. The half-filled symbols show the tetramers and the outlined symbols are the pentamers. Bar graphs b and d**
**compare total cluster concentration of a given size ($[N_m]$) between aceate (red) and nitrate (black) for the same [DMA] and**
**$[A_1]_0$ as a and b respectively.**





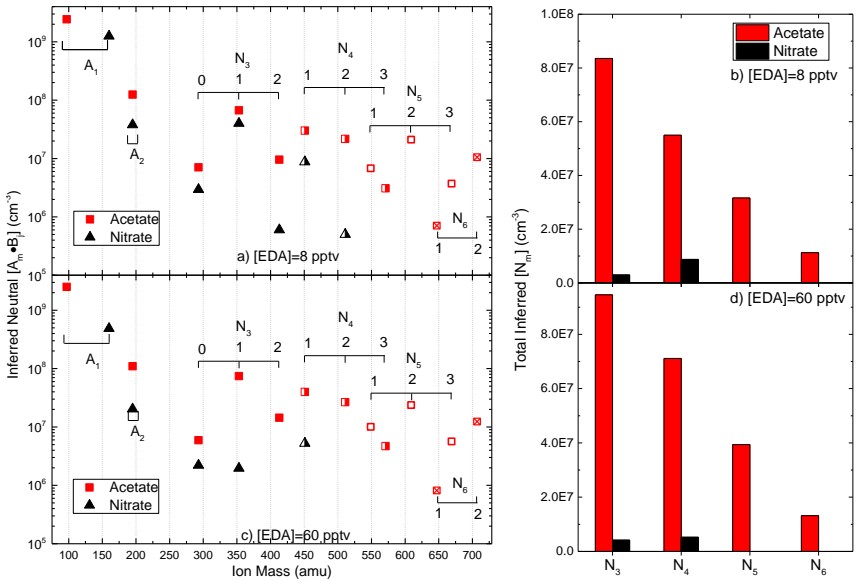

**Figure 2 (a and c) Comparison of specific cluster concentrations ([$A_m \cdot B_j$]) using acetate (red squares) and nitrate (black**
**triangles) reagent ions at two different [EDA] and constant intial sulfuric acid concentration, [$A_1$]$_o$~4x10$^9$ cm$^{-3}$. Each cluster**
**species is shown at its ion mass. The brackets represent the number of EDA molecules in a cluster with a given number of**
**sulfuric acid. The half-filled symbols show the tetramers, outlined symbols as the pentamers, and crossed symbols as 6-mer.**
**Bar graphs b and d compare total cluster concentration of a given size ([$N_m$]) between aceate (red) and nitrate (black) for**
**the same [EDA] and [$A_1$]$_o$ as a and b respectively.**

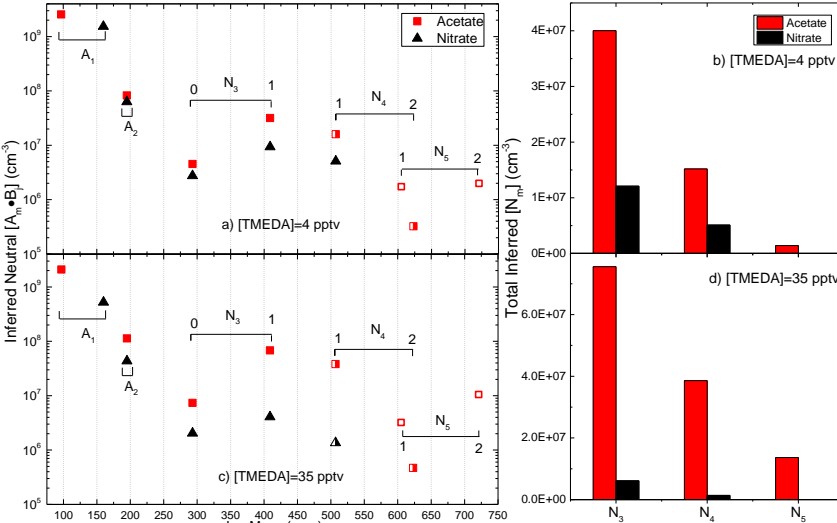


**Figure 3 (a and c) Comparison of specific cluster concentrations ([$A_m \cdot B_j$]) using acetate (red squares) and nitrate (black**
**triangles) reagent ions at two different [TMEDA] and constant intial sulfuric acid concentration, [$A_1$]$_o$~4x10$^9$ cm$^{-3}$. Each**
**cluster species is shown at its ion mass. The brackets represent the number of TMEDA molecules in a cluster with a given**
**number of sulfuric acid. The half-filled symbols show the tetramers and outlined symbols as the pentamers. Bar graphs b**
**and d compare total cluster concentration of a given size ([$N_m$]) between aceate (red) and nitrate (black) for the same**
**[TMEDA] and [$A_1$]$_o$ as a and b respectively.**



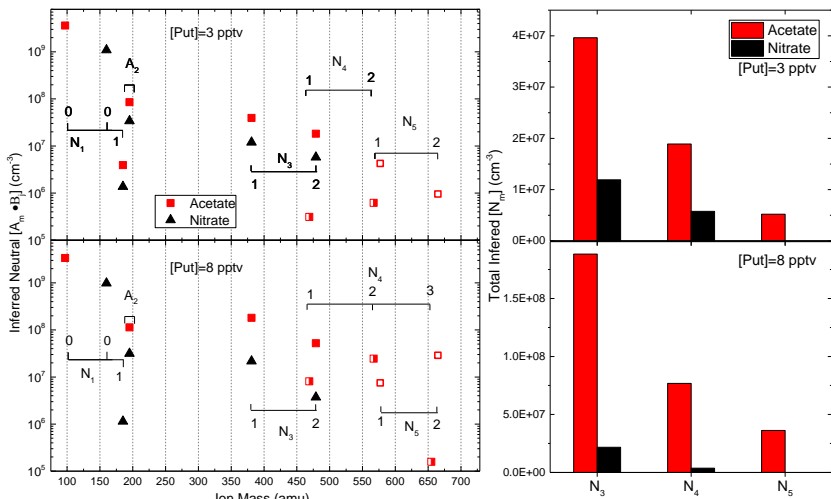


**Figure 4 (a and c)** Comparison of specific cluster concentrations ($[A_m \cdot B_j]$) using acetate (red squares) and nitrate (black triangles) reagent ions at two different [Put] and constant intial sulfuric acid concentration, $[A_1]_o \sim 4 \times 10^9$ cm$^{-3}$. Each cluster species is shown at its ion mass. The brackets represent the number of Put molecules in a cluster with a given number of sulfuric acid. The half-filled symbols show the tetramers and outlined symbols as the pentamers. Bar graphs b and d compare total cluster concentration of a given size ($[N_m]$) between aceate (red) and nitrate (black) for the same [Put] and $[A_1]_o$ as a and b respectively.

Chemical ionization efficiency clearly plays a role in both the types and amounts of clusters that can be detected. However, the concentrations in Figures 1 through 4 were calculated by assuming negligible contributions of IIC and ion decomposition. The validity of these assumptions was tested by examining the ion behavior with CI reaction time ($t_{CI}$) for a variety of bases. Presented in the following sections are ion signal variations with $t_{CI}$ and a discussion of possible scenarios that explain these observations. To help understand these measurements, we developed a model to describe these complex series of reactions that govern neutral cluster formation, chemical ionization, IIC, and ion decomposition. The model combines two box models: one for neutral cluster formation and one for the ion processes. When compared to observations, the model was useful in identifying the controlling process for the monomer and dimer but, due to the numerous reactions, only provided general scenarios to explain observations for the larger clusters.

### *Monomer, $N_1$:*

Over the 3 s neutral reaction time in this flow reactor (i.e., the reaction time between neutral sulfuric acid vapor and the basic gas), initial monomer concentration ($[N_1]$) is depleted as it forms larger clusters/particles and is lost to walls; $N_1$ may re-enter the gas phase by evaporation of larger clusters. Two types of $N_1$ may have significant abundances in the sulfuric acid and DMA system: $A_1$ and $A_1 \cdot$DMA. One computational chemistry study predicts the latter has an evaporation rate of $10^{-2}$ s$^{-1}$ (all computed rates at 298 K unless otherwise stated) (Ortega et al., 2012) with others suggesting an evaporation rate closer to 10 s$^{-1}$ (Nadykto et al., 2014;Bork et al., 2014).

Following the neutral reactions, the remaining monomer is readily chemically ionized and the product ion can decompose and undergo IIC with the monomer or clusters. For example, the decomposition rate of $A_1^- \cdot$DMA is predicted to be $10^9$ s$^{-1}$ (Ortega et al., 2014). Therefore, whether or not $A_1 \cdot$DMA is a significant fraction of the total monomer concentration, $A_1^-$ is the only ion with significant abundance. This agrees with our experimental observations.



241   Neutral $[N_1]$ can be estimated from mass spectrometry signals because there is negligible ion breakup in the
242 Cluster CIMS that leads to $A_1^-$. As discussed above, a number of experiments and the current results have shown this
243 to be the case (Hanson and Eisele, 2002;Eisele and Hanson, 2000;Lovejoy and Bianco, 2000). The signal ratio of the
244 sulfuric acid monomer at 160 amu for nitrate ($S_{160}$) to the nitrate ion at 125 amu ($S_{125}$) can be converted to neutral $[N_1]$
245 following Equation 1 (Eisele and Hanson, 2000), where $t_{CI}$ is the CI reaction time.

$$\frac{S_{160}}{S_{125}} = k_1 [N_1] t_{CI}$$
    **Equation 1**

246 For $N_1 + HNO_3 \cdot NO_3^-$, $k_1 = 1.9 \times 10^{-9}$ cm$^3$ s$^{-1}$ (Viggiano et al., 1997) which is assumed to not depend on whether water or
247 bases are attached onto the monomer. Equation 1 was derived for short $t_{CI}$ where reagent ion and neutral $N_1$ are not
248 depleted. These assumptions are tenuous at long $t_{CI}$ ; however, the rigorous analytical solution to the population
249 balance equations (derived in the SI and given in Equation S6) shows that Equation 1 is a good approximation: at
250 $t_{CI}=15$ or 18 ms, the differences between Equation 1 and Equation S6 are ~1%.

251   Figure 5 (a and b) shows the signal ratios as a function of $t_{CI}$ for DMA and EDA as detected by nitrate CI at
252 equivalent $[A_1]_o = 4 \times 10^9$ cm$^{-3}$. TMEDA and Put graphs look very similar to EDA (see SI). The green points shown in
253 this figure and subsequent figures provide measurements at base concentration of 0 pptv from eight different days and
254 offer a useful guide for the measurement uncertainty. For all base concentrations as $t_{CI}$ increases, more $[N_1]$ is
255 chemically ionized, leading to higher $S_{160}/S_{125}$. As $[B]$ increases, the signal ratios and therefore the slopes of the lines
256 decrease. This indicates that $[N_1]$ is depleted during the 3 s neutral reaction time via uptake into large clusters that
257 increase with $[B]$.

258   The model, as mentioned above, was used to interpret the results presented in Figure 5 and subsequent graphs.
259 The neutral cluster concentrations after $[A_1]_o$ and $[B]$ react over the 3 s neutral reaction time are modeled first. This
260 portion of the model also takes into account base dilution from its injection point in the flow reactor (see Jen et al.
261 (2014)), wall loss, and particle coagulation. However, the model does not take into account possible dilution of $N_1$ by
262 the base addition flow which may affect measured $[N_1]$ as explained in the SI. The neutral model is then coupled to
263 the ion model which simulates chemical ionization and IIC. Ion decomposition is implicitly included by assuming
264 certain cluster types instantly decompose into the observed ion.

265   For the monomer, the model has identical neutral cluster formation pathways for all sulfuric acid and base
266 systems. The acetate vs. nitrate comparison suggests that monomers containing various bases are chemically ionized
267 similarly, with a slight possibility that nitrate may not chemically ionize sulfuric acid monomers that contain a diamine.
268 The modeled reactions pertaining to the monomer are given in Table 1, where $k_c$ is $2 \times 10^{-9}$ cm$^3$ s$^{-1}$. The full list of
269 modeled reactions, including loss of monomer to form larger clusters, is given in the SI.



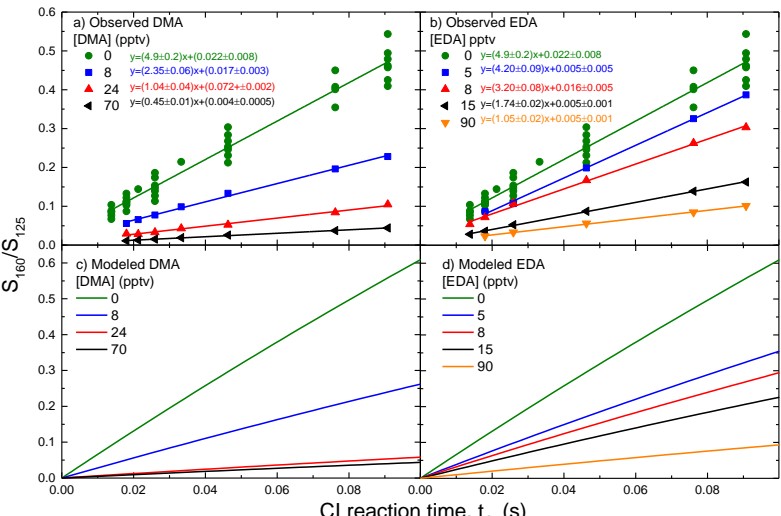

270

**Figure 5 Measured (a,b) and modeled (c, d) $S_{160}/S_{125}$ as a function of CI reaction time for DMA (a, c) and EDA (b, d). The measurements were conducted with nitrate as the reagent ion and at $[A_1]_o \sim 4 \times 10^9$ cm$^{-3}$. Each color represents a different [B] with the linear regressions for the measurements given in colored text.**

**Table 1 Summary of possible pathways for neutral monomer formation and chemical ionization**

| Neutral formation | Nitrate CI and ion decomposition |
|---|---|
| DMA and Diamines: $$A_1 + B \underset{E_1}{\overset{k}{\rightleftarrows}} A \bullet B$$ | DMA: $$A_1 + NO_3^- \xrightarrow{k_c} HNO_3 \bullet A_1^-$$ $$A_1 \bullet B + NO_3^- \xrightarrow{k_c} HNO_3 \bullet A_1^- \bullet B$$ $$HNO_3 \bullet A_1^- \bullet B \xrightarrow{fast} HNO_3 \bullet A_1^- + B$$ Diamines: $$A_1 + NO_3^- \xrightarrow{k_c} HNO_3 \bullet A_1^-$$ $$A_1 \bullet B + NO_3^- \xrightarrow{?} HNO_3 \bullet A_1^- \bullet B$$ $$HNO_3 \bullet A_1^- \bullet B \rightarrow HNO_3 \bullet A_1^- + B$$ |

Figure 5 (c and d) displays the modeled results for DMA and EDA at the same [B] and $[A_1]_o$ as the
measurements presented in panels a and b. The model predicts the linear dependence of $S_{160}/S_{125}$ on $t_{CI}$ as seen in
Equation 1. In addition, the predicted values of $S_{160}/S_{125}$ and their dependence on [B] are in good qualitative agreement
with observations. Including or excluding nitrate CI of $A_1 \bullet$diamine has little effect on $S_{160}/S_{125}$ because [B] is typically
less than $[A_1]_o$ in these experiments. As a result, the majority of monomers will remain as $A_1$ even if the evaporation
rate of the $A_1 \bullet B$ ($E_1$) is very small. Further experiments that quantify the fraction of $A_1 \bullet$diamine in $N_1$ are needed to
definitely conclude the efficacy of nitrate in chemically ionizing all $N_1$.
**Dimer, $N_2$:**
Neutral dimers ($N_2$) largely form by collision of the two types of monomers ($A_1$ and $A_1 \bullet B$) and, to a much
lesser extent, decomposition of larger clusters. For sulfuric acid+DMA, the $N_2$ likely exists as $A_2 \bullet$DMA and $A_2 \bullet$DMA$_2$,
with both clusters predicted to have low evaporation rates of $\sim 10^{-5}$ s$^{-1}$ (Ortega et al., 2012) with another study





suggesting a higher evaporation rate of $A_2 \cdot DMA_2$ ~$10^4$ times higher (Leverentz et al., 2013). Chemically ionizing
these dimers results in ions that undergo IIC and ion decomposition. Computational chemistry predicts that $A_2^- \cdot DMA_2$
and $A_2^- \cdot DMA$ have DMA evaporation rates of $10^8$ s$^{-1}$ and $10^2$ s$^{-1}$, respectively (Ortega et al., 2014). However, the
computed evaporation rate of $A_2^- \cdot DMA$ may be too low because during the 18 ms CI reaction time used here, all $N_2$
are detected as $A_2^-$ (195 amu). Similarly, the diamine molecule is lost from $A_2^- \cdot$diamine as all dimers were detected as
$A_2^-$.
$A_2^-$ can also be created from IIC between $A_1^-$ and $N_1$ (see Reaction 2) that proceeds with a rate coefficient of
$k_{21}$. Including both processes in the cluster balance equations leads to the ratio of sulfuric acid dimer (195 amu) to
monomer (160 amu) signal intensities shown in Equation 2. This relationship includes a time-independent term (the
$t_{CI}$=0 s intercept) that is proportional to the neutral dimer to monomer ratio in the sampled gas, and a term due to IIC
that increases linearly with $t_{CI}$ (Chen et al., 2012;Hanson and Eisele, 2002).

$$\frac{S_{195}}{S_{160}} = \frac{k_2}{k_1}\frac{[N_2]}{[N_1]} + \frac{1}{2}k_{21}[N_1]t_{CI}$$

**Equation 2**

The rate constants, $k_{ij}$, are the collisional rate constants. Equation 2 was also derived from the assumption of short $t_{IC}$.
The relation for $S_{195}/S_{160}$ vs. $t_{CI}$ for long $t_{CI}$ is also derived in the SI. Equation 2 is a good approximation for the more
rigorous solution even at long $t_{IC}$.
Figure 6 (a-c) shows measured $S_{195}/S_{160}$ as a function of $t_{CI}$ for DMA, EDA, and TMEDA respectively as
detected by nitrate CI at $[A_1]_o$=4x10$^9$ cm$^{-3}$. Put is similar to EDA and is presented in Figure 7 (left). For all bases,
increasing the CI reaction time leads to more IIC-dimer. The observed linear increase in the $S_{195}/S_{160}$ ratio for all bases
provides evidence for the influence of IIC on dimer measurements (Equation 2). However, the y-intercepts for DMA
exhibit a pattern that is distinctly different from those observed for the diamines, indicating different trends for the
neutral monomer to dimer concentration ratios. For DMA, the y-intercept increases with increasing [B]. This is due
to higher concentrations of base depleting the monomer and enhancing dimer concentrations. A different trend was
observed for the diamines with the intercepts showing no clear dependence on diamine concentration.




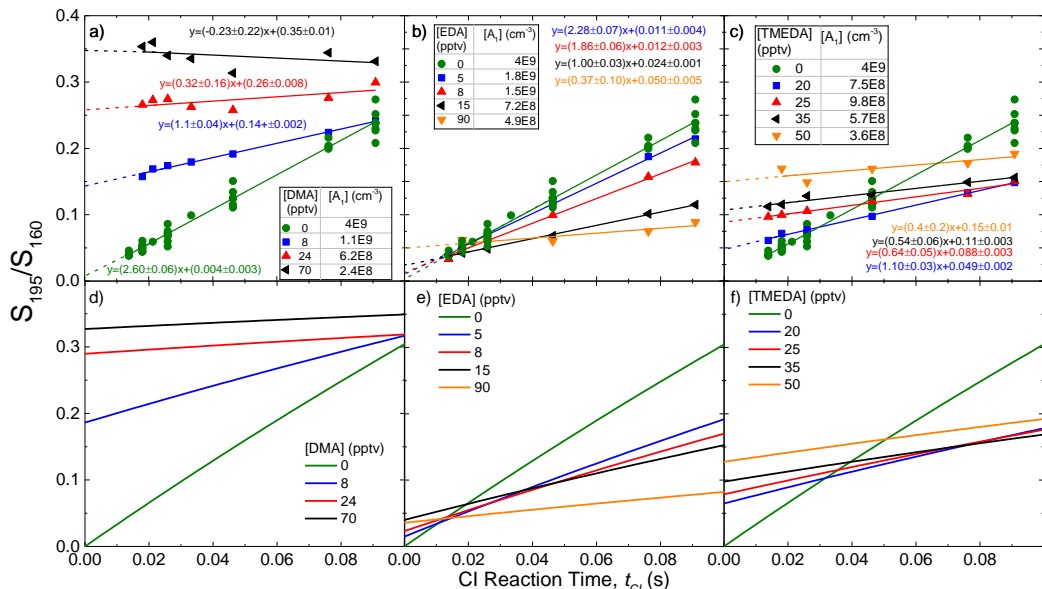


**Figure 6 Measured $S_{195}/S_{160}$ as a function of $t_{CI}$ for DMA (a), EDA (b), and TMEDA (c) measured by nitrate CI at $[A_1]_o \sim 4 \times 10^9$ $cm^{-3}$. The tables in panels a-c provide the measured $[A_1]$ at that $[B]$. Observations were fitted according to Equation 2 with the y-intercept shown by the dashed line. Panels d-f present modelled results for each base.**

There are a number of scenarios that could partly explain the diamine trends. First, the neutral trimer evaporation rate(s) could be very low such that the formation of trimer and larger clusters will deplete both $[N_2]$ and $[N_1]$. $A_1$ evaporation rate from $A_3 \cdot DMA$ is predicted to be $\sim 1$ $s^{-1}$ (Ortega et al., 2012) and likely lower for cluster with diamines (Jen et al., 2016). The second possibility is $A_2^-$ could be the decomposition product of larger ions such as $A_3^- \cdot$ diamine forming $A_2^- + A_1 \cdot$ diamine. A third possibility is that $A_2 \cdot$ diamine$_2$ cannot be readily ionized by nitrate as compared to $A_2 \cdot DMA_2$ possibly due to differences in cluster configurations and dipole moments. As [diamine] increases, the fraction of dimers containing two diamines increases, resulting in a growing fraction of $N_2$ that may not be ionizable by nitrate. For example, the model predicts $[A_2 \cdot EDA]$ is 10% of $[A_2 \cdot EDA_2]$ when $[EDA]$=90 pptv.

The dimer ($S_{195}$) to monomer signal ($S_{97}$) ratio for sulfuric acid+Put dimers measured using acetate CI as a function of $t_{CI}$ was examined to better understand which of these explanations is the most relevant. As mentioned previously, acetate detects the sulfuric acid monomer as 97 amu, but the detected dimer is at 195 amu for both nitrate and acetate. Figure 7 shows the ratio of these signals for Put between nitrate (a) and acetate (c). At [Put]=40 pptv, acetate shows a $S_{195}/S_{97}$ y-intercept 25 times higher than the intercepts shown in the nitrate graph. The higher y-intercepts are most likely due to improved CI efficiency. Decreased detection efficiency of 97 amu and an increased contribution due to $A_3^- \cdot$ diamine decomposition due to better CI of $N_3$ by acetate may also contribute (although high $[A_3^- \cdot$ diamine] in Figure 4 suggests these ions are stable enough during the acetate $t_{CI}$=15 ms). More acetate results similar to Figure 7 (c) are needed to draw a more definitive conclusion, but these comparisons do suggest that dimers containing 1-2 diamines are not inefficiently chemically ionized by nitrate in these experiments.




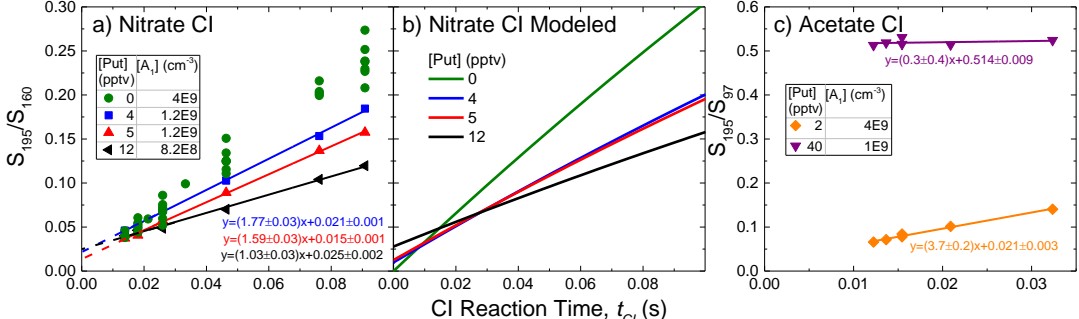

**Figure 7 Measured dimer to monomer signal ratios as a function of CI reaction time using nitrate (a) and acetate CI (c). In both cases, $[A_1]_o$ was held constant at $4 \times 10^9$ cm$^{-3}$. Panel (b) shows the modeled results for Put.**

The model adds more clarity on why $N_2$ containing diamines behave differently than DMA using nitrate CI. For DMA, the best fit to the observations was achieved by assuming all clusters can undergo nitrate ionization and can be formed by IIC. In addition, base evaporation rates from $A_2 \cdot B_2$ and sulfuric acid evaporation rates from the trimer were set to 0 s$^{-1}$; increasing these evaporation rates (up to 10 and 5 s$^{-1}$ respectively) had little effect on the ratio trends. The model also assumed that $A_3^- \cdot B$ does not decompose into $A_2^-$. Figure 6 (d) shows modeling results for DMA. To reproduce $S_{195}/S_{160}$ trends of EDA and Put, the model followed that of DMA except $A_2 \cdot B_2$ cannot be ionized by nitrate. For TMEDA, the model also assumed $A_2 \cdot TMEDA_2$ does not form. Modeled results are shown in Figure 6 (e and f) for EDA and TMEDA, respectively, and Figure 7 (b) for Put. The modeled pathways for $N_2$ are listed in Table 2. Several of the modeled reactions are simplified versions of multi-step reactions. For example, preventing the formation of $A_2 \cdot TMEDA_2$ could also mean $A_2 \cdot TMEDA_2$ forms at the collision rate but instantly decomposes into $A_2 \cdot TMEDA$. For all three diamines, we were unable to reproduce the observations with other combinations of reactions and evaporation rates. The model only matched the observed trends when turning off the CI or formation of $A_2 \cdot diamine_2$.

Other explanations may exist to explain the differences between DMA and diamines observations (the most likely being semi-efficient CI of $A_2 \cdot diamine$ instead of zero nitrate CI of $A_2 \cdot diamine_2$), but additional thermochemical data (e.g., from more targeted experiments and computational chemistry) are needed to better inform the model. Regardless, our observations and modeling show that dimer's neutral formation pathways and/or the nitrate CI differs between the DMA and diamine systems.

The model also provides an estimate of the fraction of $[A_2^-]$ formed by IIC at $t_{CI}$=18 ms (used for the nitrate CI experiments). For base concentration of 0 pptv, the model is very similar to what was measured in Figure 6, indicating that $A_2^-$ is almost completely formed by $A_1^- + A_1$ (i.e., is an IIC artifact) and not by the CI of $A_2$. The abundance of $A_2$ is low at 300 K (Hanson and Lovejoy, 2006), below detection limit of the Cluster CIM. For DMA, IIC dimers typically account for 1% (less at high [DMA]) of the total dimer signal which agrees with the conclusions drawn in Jen et al. (2015). In contrast, the IIC fraction of $A_2^-$ using nitrate for EDA and Put is ~50%, due to the potentially large fraction $N_2$ not undergoing chemical ionization. The nitrate ion's inability to chemically ionize some of the dimers is further highlighted since IIC is suppressed in the diamine system: less $N_1$ is available (due to formation of larger clusters) thus both $[A_1]$ and $[A_1^-]$ are depressed. IIC-produced $A_2^-$ accounts for ~20% of the total dimer signal for TMEDA. However, these numbers are uncertain due to the assumptions in the model and uncertainties in the measurement. For instance, the model is not sensitive to whether $A_1^-$ can cluster with $A_1 \cdot B$, which would significantly influence the amount of IIC dimer without significantly affecting $S_{195}/S_{160}$. IIC contributes much less $A_2^-$ when acetate is used as the reagent ion because acetate detects up to 5 times more total neutral dimer concentration ($[N_2]$) than nitrate when base is present. Acetate measurements show that IIC produced ~3% of the $[A_2^-]$ when [Put]=2 pptv and near zero when [Put]=40 pptv (Figure 7 c).





**Table 2 Summary of possible pathways for neutral and ion dimer formation**

| Neutral formation | Nitrate CI and ion decomposition reactions | IIC reactions (only $A_1^-$) |
|---|---|---|
| DMA, Put, EDA: $$A_1 \bullet B + A_1 \xrightarrow{k} A_2 \bullet B$$ $$A_1 \bullet B + A_1 \bullet B \xrightarrow{k} A_2 \bullet B_2$$ $$A_2 \bullet B + B \xrightarrow{k} A_2 \bullet B_2$$ $$A_2 \bullet B_2 \xrightarrow{E_{2B}} A_2 \bullet B + B$$ TMEDA: $$A_1 \bullet B + A_1 \xrightarrow{k} A_2 \bullet B$$ $$A_1 \bullet B + A_1 \bullet B \xcancel{\longrightarrow} A_2 \bullet B_2$$ $$A_2 \bullet B + B \xcancel{\longrightarrow} A_2 \bullet B_2$$ | DMA: $$A_2 \bullet B + NO_3^- \xrightarrow{k_c} A_2^- \bullet B + HNO_3$$ $$A_2^- \bullet B \xrightarrow{fast} A_2^- + B$$ $$A_2 \bullet B_2 + NO_3^- \xrightarrow{k_c} A_2 \bullet B_2^- + HNO_3$$ $$A_2^- \bullet B_2 \xrightarrow{fast} A_2^- \bullet B$$ Diamines: $$A_2 \bullet B + NO_3^- \xrightarrow{k_c} A_2^- \bullet B$$ $$A_2^- \bullet B \xrightarrow{fast} A_2^- + B$$ $$A_2 \bullet B_2 + NO_3^- \xcancel{\longrightarrow} A_2^- \bullet B_2$$ | All bases: $$A_1^- + A_1 \xrightarrow{k_c} A_2^-$$ $$A_1^- + A_1 \bullet B \xrightarrow{k_c} A_2^- \bullet B$$ |


**Trimer, $N_3$:**

Neutral trimers ($N_3$) are primarily formed by combining one of the two types of monomers with one of the
two types of dimers; evaporation of large clusters also contributes. In the sulfuric acid+DMA system, computational
chemistry predicts $A_3 \bullet DMA_2$ and $A_3 \bullet DMA_3$ are relatively stable, with $A_3 \bullet DMA_3$ exhibiting the lowest evaporation
rate (Ortega et al., 2012). Also $A_3 \bullet DMA$ may be present in significant amounts due to a high production rate via
$A_2 \bullet DMA + A_1$. CI of $N_3$ leads to ions such as (i) $A_3^- \bullet DMA_3$ which evaporate at a rate of $10^4$ s$^{-1}$ into $A_3^- \bullet DMA_2$ and (ii)
$A_3^- \bullet DMA_2$ and $A_3^- \bullet DMA$ which have predicted evaporation rates of ~$10^{-1}$ and $10^{-2}$ s$^{-1}$ (Ortega et al., 2014),
respectively, resulting in lifetimes comparable to $t_{CI}$ used here. From Figure 1, nitrate CI resulted in $A_3^- \bullet DMA_2$ (only
at [DMA]=110 pptv), $A_3^- \bullet DMA$, and $A_3^-$. The DMA-containing clusters were detected to a much lesser extent than
with acetate CI.
Acetate CI results help shed light on these processes with much higher [$A_3^- \bullet DMA_{1,2}$] than with nitrate CI
(Figure 1) which could be due to decomposition of larger ion clusters. The acetate CI results depicted in Figure 1 show
that $A_3^- \bullet DMA_2$ is the most abundant type of trimer ion, suggesting that the dominant neutral clusters are $A_3 \bullet DMA_{2-3}$,
with any $A_3^- \bullet DMA_3$ quickly decomposing into $A_3^- \bullet DMA_2$. Neutral $A_3 \bullet DMA_3$ is predicted by our model to be dominant
at high [DMA]. This picture is consistent with our postulate that nitrate cannot ionize $A_3 \bullet DMA_3$ (and also, possibly,
$A_3 \bullet DMA_2$) and thus little $A_3^- \bullet DMA_{1,2}$ is observed using nitrate CI.
The trimer ions observed using acetate CI may have contributions from decomposition of large clusters. For
example, $A_3^- \bullet DMA_2$ could be formed by the decomposition of $A_4^- \bullet DMA_2$ or $A_4^- \bullet DMA_3$ via loss of $A_1$ or $A_1 \bullet DMA$,
respectively. If these types of processes are significant, they might explain some of the differences in the trimer ion
observations between nitrate and acetate CI. Highly aminated tetramer neutrals would be more readily ionized by
acetate and result in larger contributions to the trimer ion signals than compared nitrate CI. Thus, this may be one
drawback to acetate CI: a possible shift downwards in sulfuric acid content in the distribution of ions vs. the neutrals.
The sulfuric acid + diamine system shows nitrate CI detection of $A_3^- \bullet diamine_{0-2}$ but at much lower
abundances than acetate CI, particularly for EDA. Interestingly, the most abundant trimer ions after acetate CI contain
on average 1 diamine molecule compared to 2 in the DMA system. This is consistent with particle measurements that
show one diamine molecule is able to stabilize several sulfuric acid molecules, and thus form a stable particle, while
at least 2 DMA molecules are required for the same effect (Jen et al., 2016). The two amino groups on the diamine





molecule can both effectively stabilize trimers, and this size is stable for the relevant time scales in this flow reactor
(Glasoe et al., 2015;Jen et al., 2016). Therefore, larger clusters can be produced with higher acid to base ratios.

To better understand the trimer ion behaviors, we monitored the bare trimer signal ($A_3^-$, $S_{293}$) and monomer
signal ($S_{160}$) as a function of CI reaction time, $t_{CI}$. Figure 8 shows $S_{293}/S_{160}$ for nitrate CI for DMA, EDA, and TMEDA
at $[A_1]_0 = 4 \times 10^9$ cm$^{-3}$. Note, equivalent measurements for Put are similar to those of EDA. Low values of $S_{293}/S_{160}$ for
all conditions indicate minimal creation of $A_3^-$ from the CI of $N_3$. Thus, IIC-produced $A_3^-$ can be a significant fraction
of observed $A_3^-$. Without base present, IIC is the only way to produce detectable amounts of $A_3^-$ (green circles in
Figure 8).

$A_3^-$ can also be formed by the decomposition of larger ions such as $A_3^- \cdot B$. Evidence of this decomposition
can be seen in Figure 9 where $S_{A3 \cdot B}/S_{160}$ measured using nitrate CI is shown as a function of $t_{CI}$. For diamines at high
concentrations and short $t_{CI}$, $S_{A3 \cdot B}/S_{160}$ decreases with $t_{CI}$ and can be attributed to decomposition of this ion. Shorter
$t_{CI}$ allows the instrument to capture short-lived ions. $A_3^- \cdot$diamine decomposes at longer times and could form $A_3^-$,
thereby decreasing $S_{A3 \cdot B}/S_{160}$ and increasing $S_{293}/S_{160}$. However, $S_{293}/S_{160}$ for the diamines does not increase with $t_{CI}$,
indicating that $A_3^- \cdot$diamine likely decomposes into products other than $A_3^-$. The DMA system also exhibits a very
small decrease of $S_{A3 \cdot B}/S_{160}$ at short $t_{CI}$, but ratio values are within measurement uncertainties. Thus no conclusion
can be drawn from this decrease of $S_{A3 \cdot DMA}/S_{160}$ at short $t_{CI}$.

Another, more likely scenario to explain these time dependent behaviors for the trimer ion signals is if $A_3^- \cdot B$
decays into $A_2^-$ and a neutral $A_1 B$ at short $t_{CI}$. Assuming we have captured most of the initial $A_3^- \cdot B$ signal at the
shortest $t_{CI} = 15$ ms in Figure 9 (a-c), the increase in $A_2^-$ due to this mechanism would be small compared to the observed
$A_2^-$ signal. Acetate data for Put (Figure 7 c) provide some evidence supporting this because the slope of the [Put]=2
pptv is 3.7 and is higher than the 2.6 slope of [B]=0 pptv case. Since $A_2^-$ when [B]=0 pptv is primarily produced by
IIC, a higher slope when [Put]=2 pptv indicates larger ion decomposition contributing to the $A_2^-$ signal.

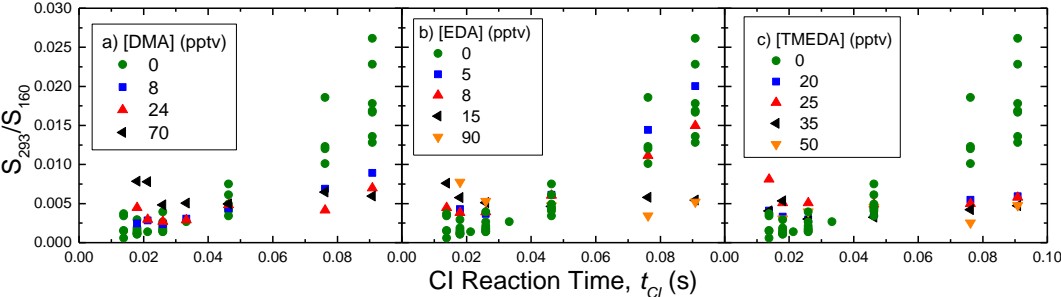

**Figure 8 Measured $S_{293}/S_{160}$ as a function of $t_{CI}$ for DMA (a), EDA (b), and TMEDA (c) detected by nitrate CI at $[A_1]_0 = 4 \times 10^9$**
**cm$^{-3}$.**





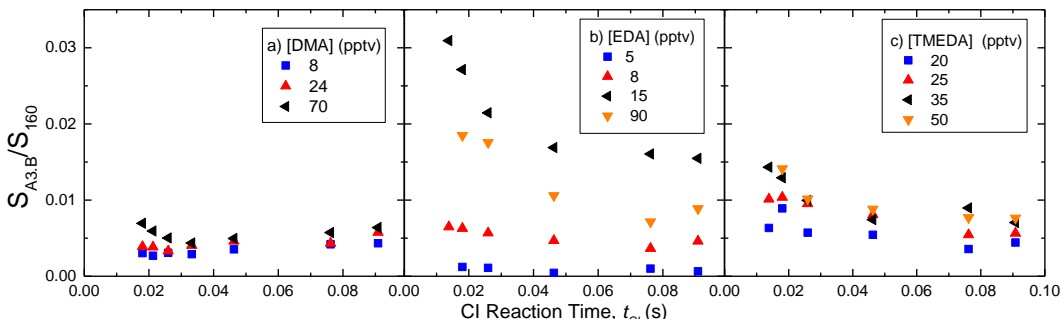

**Figure 9 Nitrate measured $S_{A3 \cdot B}/S_{160}$ as a function of $t_{CI}$ for DMA (a), EDA (b), and TMEDA (c) at $[A_1]_0 = 4 \times 10^9$ cm$^{-3}$. Note**
**the different y-axis scales between bases.**

Scenarios deduced from these trimer ion observations and previous computational chemistry studies for the
sulfuric acid and DMA system are summarized in Table 3. These reactions have little effect on the modeled dimer
results since they introduce minor dimer ion sources. In contrast, each trimer pathway adds large uncertainty to the
modeled trimer behavior. For example, including ion decomposition reactions of larger ions (tetramer and larger),
postulated from the acetate CI results, may greatly influence concentration of smaller trimer ions which already exhibit
very low signals using nitrate CI. In addition, nitrate inefficient ionization of neutral trimers leads to large uncertainties
in modeling the unobserved trimer types. More detailed observations of the chemically neutral trimers and
computational chemistry studies on evaporation rates for sulfuric acid+diamine systems will improve future efforts to
model these processes.

**Table 3 Summary of possible pathways for neutral and ion trimers formed from sulfuric acid and DMA, excluding**
**decomposition of tetramer and larger ions**

| Neutral formation | Nitrate CI and ion decomposition reactions | IIC reactions (only $A_1^-$) |
|---|---|---|
| $A_2 \bullet B + A_1 \xrightarrow{k} A_3 \bullet B$ | $A_3 \bullet B + NO_3^- \xrightarrow{k_c} A_3^- \bullet B + HNO_3$ | $A_2^- + A_1 \xrightarrow{k_c} A_3^-$ |
| $A_3 \bullet B + B \xrightarrow{k} A_3 \bullet B_2$ | $A_3^- \bullet B \xrightarrow{E_d} A_2^- + A_1 \bullet B$ | $A_1^- + A_2 \bullet B \xrightarrow{k_c} A_3^- \bullet B$ |
| $A_2 \bullet B_2 + A_1 \xrightarrow{k} A_3 \bullet B_2$ | $A_3 \bullet B_3 + NO_3^- \not\longrightarrow A_3^- \bullet B_3 + HNO_3$ | |
| $A_2 \bullet B + A_1 \bullet B \xrightarrow{k} A_3 \bullet B_2$ | $A_3 \bullet B_2 + NO_3^- \not\longrightarrow A_3^- \bullet B_2 + HNO_3$ | |
| $A_2 \bullet B_2 + A_1 \bullet B \xrightarrow{k} A_3 \bullet B_3$ | | |


**Tetramer, $N_4$:**
Nitrate CI leads to very low amounts of tetramer ions and primarily as $A_4^- \bullet DMA_{1-3}$ and $A_4^- \bullet diamine_{1,2}$.
Computational chemistry suggests that the sulfuric acid+DMA tetramer likely exists as $A_4 \bullet DMA_{2-4}$, with $A_4 \bullet DMA_4$
dominating the population (Ortega et al., 2012). The acetate data appears to confirm this with $A_4^- \bullet DMA_3$ as the most
abundant tetramer ion which likely predominately originated from the decomposition of $A_4 \bullet DMA_4$ upon ionization
(Ortega et al., 2014). Nitrate may efficiently chemically ionize $A_4 \bullet DMA_{1-2}$, however their concentrations after the 3 s
neutral reaction time are likely below the detection limit of the Cluster CIMS ($<10^5$ cm$^{-3}$). Furthermore, the $A_4^-$
$\bullet DMA_{1,2}$ ions may be subject to elimination of $A_1 \bullet DMA$. Nitrate CI results show ~100 times higher $[A_4^- \bullet diamine]$
than $[A_4^- \bullet DMA]$ at about equivalent initial reactant concentrations. This suggests that the most stable neutral tetramers
contain fewer diamine molecules than DMA. In addition, the acetate CI results for the diamines show the majority of



N$_4$ contain 1 diamine, further supporting the conclusions drawn in Jen et al. (2016) that only one diamine molecule is
needed to form a stable particle.
Due to the very low observed concentration of A$_4^-$•DMA, we focus on the ions of the diamine systems. The
stability and behavior of A$_4^-$•diamine can be examined by looking at nitrate detected signal ratios of A$_4^-$•diamine and the
monomer ($S_{A4•diamine}/S_{160}$) as a function of CI reaction time, given in Figure 10. Similar to A$_3^-$•EDA, $S_{A4•EDA}/S_{160}$ and
$S_{A4•Put}/S_{160}$ decreases with time at short $t_{CI}$, indicating that they decompose with a lifetime shorter than a few tens of
ms. $S_{A4•TMEDA}/S_{160}$ also shows a decrease at short $t_{CI}$, but it is less evident. It could have a fast decay rate leading to a
few ms lifetime, and our measurements would have mostly missed them. Nonetheless, decomposition of A$_4^-$•diamine
likely entails evaporation of N$_1$ or N$_2$ instead of a lone diamine from the cluster as [A$_4^-$] was below detection limit of
the Cluster CIMS using nitrate. At long CI reaction time, $S_{A4•EDA}/S_{160}$ remained constant, indicating negligible
contribution of IIC to A$_4^-$•EDA signal. In contrast, $S_{A4•Put}/S_{160}$ and $S_{A4•TMEDA}/S_{160}$ increase at long $t_{CI}$. This could be
due to IIC or larger ion decomposition.

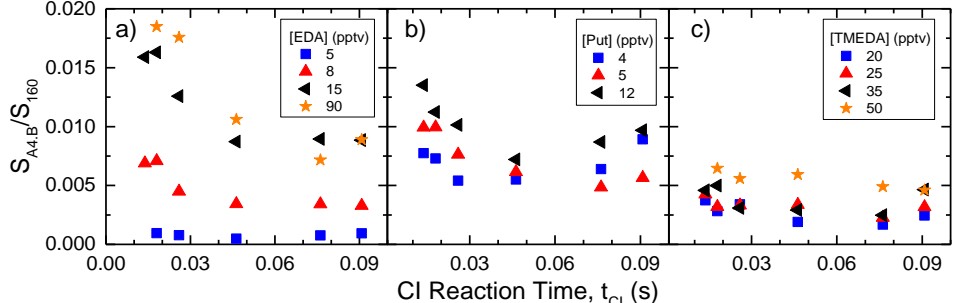

**Figure 10 Nitrate measured $S_{A4.diamine}/S_{160}$ as a function of CI reaction time for EDA (a), Put (b), and TMEDA (c).**
**Pentamer, N$_5$:**
Nitrate CI did not detect any pentamer (N$_5$), but pentamer was detected using acetate CI. In the diamine
system, acetate detected N$_5$ with fewer diamine molecules (1-2) than DMA (4). However, A$_5^-$•EDA$_{>3}$, A$_5^-$•TMEDA$_{>1}$,
and A$_5^-$•Put$_{>2}$ fall outside the Cluster CIMS mass range of 710 amu. Thus, we may not have measured the complete
pentamer population. The most abundant N$_5$ is A$_5^-$•DMA$_4$ and it increases in both concentration and in fraction of N$_5$
population with increasing [DMA]. This ion could be the result of the loss of a DMA molecule after CI of A$_5$•DMA$_5$.
This would follow similar trends predicted by computation chemistry for smaller clusters. However, since
[DMA]<<[A$_1$]$_o$ (i.e., [B]/[A$_1$]$_o$ is high) and stable particles need ~2 DMA to form (Glasoe et al., 2015), [A$_5$•DMA$_5$]
as high as 10$^7$ cm$^{-3}$ would not be expected. The presence of A$_5^-$•DMA$_4$ could also then be the result of large ion
decomposition via evaporation of A$_1$ or A$_1$•DMA. Measurements of ions larger than 700 amu are needed to better
understand how they evaporate upon acetate CI and what fraction of the pentamers are not ionizable by nitrate.
**Conclusion:**
This study presents measurements of the behavior of neutral and ionized sulfuric acid clusters containing
various bases. The results show the complexities of the coupled neutral cluster formation pathways with the ion
processes (e.g. chemical ionization, ion-induced clustering, and ion decomposition). We provide various scenarios to
describe the observed trends. Our most definitive conclusions are
1)   Nitrate very likely does not chemically ionize all types of sulfuric acid dimers containing diamines. The model
indicates A$_2$•diamine$_2$ cannot be chemically ionized by nitrate. However, the model did not consider semi-efficient
nitrate CI of A$_2$•diamine which could also explain our observations.



2) Nitrate only chemically ionizes a small fraction of trimer and larger clusters in both the DMA and diamine with sulfuric acid systems. Measurements suggest that the more chemically neutral clusters are not chemically ionized by nitrate but are by acetate.

3) Acetate and nitrate CI measurements of sulfuric acid+DMA clusters generally agree with the qualitative trends of neutral and ion cluster predicted from computational chemistry (Ortega et al., 2012;Ortega et al., 2014). However, these measurements suggest that $A_3^- \bullet B$ decomposes into $A_2^-$ and $A_1 \bullet B$.

4) Nitrate measurements of $A_3^- \bullet B$ and $A_4^- \bullet B$ show that these ions decompose at roughly the same time scales as the CI reaction time at room temperature. In principle, ionization of neutral clusters leads to potentially large artifacts even before they are sampled into a vacuum system. These decomposition reactions will likely affect the calculated concentrations of the neutral clusters.

5) In an acid-rich environment where $[B]/[A_1]<1$, $A_2^-$ and $A_3^-$ are primarily produced via IIC pathways and contribute negligible amounts to overall dimer and trimer signals when any of these bases are present and at our 18 ms CI reaction time. If some fraction of the dimer is not chemically ionized by nitrate, then IIC-produced $A_2^-$ is a significant fraction of the dimer signal.

Additional computed neutral and ion evaporation rates and a more complex model combined with multivariable parameter fitting would provide more clarity to these results. In addition, more acetate CI measurements of ion signal ratios as a function of CI reaction time are needed to provide more details on specific ion behaviors. However, measurements using the acetate ion (which includes acetate, acetate•water, and acetic acid•acetate) exhibit high backgrounds in the low masses, leading to up to a factor of 5 uncertainty in measured monomer concentration ($[N_1]$) and a factor of 2-3 for dimer concentration ($[N_2]$). A higher resolution mass spectrometer is needed to resolve the background signals and reduce the uncertainties.

**Acknowledgements:**
Support from NSF Awards AGS1068201, AGS1338706, and AGS0943721 is gratefully acknowledged. CNJ acknowledges support from NSF GRFP award 00006595, UMN DDF, and NSF AGS Postdoctoral Fellowship award 1524211. J. Z. acknowledges support from SYSU 100 talents program.





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
