# Peer review of "Chemical ionization of clusters formed from sulfuric acid and dimethylamine or diamines"

_Atmospheric Chemistry and Physics, 2016_

## Referee Comment (RC1) · Anonymous Referee #1 · 15 Jul 2016

The study by Jen et al. evaluates the capabilities of two different primary ions (nitrate and acetate) in ionizing clusters composed of sulfuric acid and dimethylamine (DMA), ethylene diamine (EDA), tetramethylethylene (TMEDA), or butanediamine/putrescine (PUT). Such clusters could in principle explain atmospheric new particle formation (NPF) since the produced neutral clusters have low evaporation rates.

The neutral clusters were formed in a flow reactor at ~300 K and at 30% relative humidity. A chemical ionization mass spectrometer (cluster CIMS) was used to detect the clusters after they reacted with nitrate or acetate primary ions used for the chemical ionization. Since the formed neutral clusters can contain equal numbers of acid and base molecules their reduced acidity could make them less susceptible towards ionization by nitrate in comparison to acetate primary ions.

Indeed, the presented results indicate that some clusters can very likely not be ionized efficiently by nitrate (e.g. the sulfuric acid dimer containing two diamines, or the trimer containing DMA or diamines).

In atmospheric studies nitrate chemical ionization is generally used for measuring sulfuric acid and its influence on NPF. If some atmospheric NPF is due to sulfuric acid and amines or diamines its importance could be significantly underestimated because the absence of sulfuric acid clusters would not necessarily indicate that sulfuric acid-amine NPF is not proceeding.

For this reason the manuscript by Jen et al. is a very important contribution and will help the interpretation of the mass spectra obtained in ambient and chamber CIMS measurements. The paper is very well-written and I have no serious comments. I therefore recommend its publication after addressing the points listed in the following:

Line 31: add a space before the bracket and also between the references (after the semicolon, please check the whole manuscript)

Line 145: opening bracket is missing before $A_m.B_2$

Line 171: "factor of 2 below", does this mean all concentrations are upper estimates?

Figure 2: By comparing the panels a) and b) for the trimer it is not clear why the trimer in panel b) is >10 times higher for acetate than for nitrate. The trimer signals in panel a) are dominated by the cluster containing 3 acids and 1 base, however these signals seem to be quite similar for acetate and nitrate.

Figure 4: This figure seems to show signals for $A^-.Put$, which is surprising given the fact that in the DMA system, $A^-.DMA$ evaporates very rapidly. It would be good to include some discussion about the presence of $A^-.Put$.

Line 249 and SI equation S6: Equation S6 includes the difference between $k_{21}$ (reaction between $H_2SO_4$ and $HSO_4^-$) and $k_1$ (reaction between $H_2SO_4$ and $NO_3^-$) in the denominator of the equation. It seems that these values are identical ($2x10^{-9}$ $cm^3$ $s^{-1}$), therefore this would lead to a zero division. Please clarify.

Line 268 and line 246: two (slightly) different values for the ion-molecule reaction rates are given here. I would recommend to use the same value in the model and in equation 1.

Line 293: please provide the value of the rate constant $k_{21}$ here; this should be done for all rates by including their values in parentheses

Line 329: "efficiently" instead of "inefficiently"?

Figure 7: The figure caption states a value of $4 \times 10^9$ cm$^{-3}$ for the initial sulfuric acid concentration. However, in the legends different values for $[A_1]$ are given. Are these the concentrations after the 3 s reaction time? If so, please mention this in the figure caption.

Line 374: Do these evaporation rates refer to the evaporation of DMA or A?

Line 425: better use "sources of dimer ions" than "dimer ion sources"

Table 3: for the neutral pathway it seems the reaction $A_3B_2 + B$ is missing

Line 466: do you mean "low" instead of "high"?

References: please use same style for all references, i.e. remove hyperlinks and add page numbers for all references, etc.

SI, Line 39: "cm$^{-3}$" instead of "cm$^3$", also better to use "s$^{-1}$" instead of "Hz"

SI, Equation S6: see comment above

Table S1: In a previous paper by the same authors (Jen et al., 2016, GRL) it was concluded that diamines are a more potent source of new particles in comparison with DMA. However, from the evaporation rates listed in Table 1 this conclusion does not seem to be supported due to the rather high evaporation rates $E_1$ for the diamines (50 times higher than for DMA) and the crucial role of $A_1B_1$ in terms of cluster formation. How can this discrepancy be explained and how does it affect the conclusions from the present paper?

---

## Referee Comment (RC2) · Anonymous Referee #3 · 26 Jul 2016

Review of Jen et al., Chemical ionization of clusters formed from sulfuric acid and dimethylamine or diamines

Summary and General Comments: The authors present a series of experiments designed to assess the utility of nitrate ion CIMS techniques for the detection of H2SO4base clusters that lead to the formation of new particles in the atmosphere. Nitrate ion CIMS has been used for detection of low vapor pressure trace gases previously, and is well known to be a highly sensitive, but very specific reagent ion. The authors extend this logic to the detection of clusters, under the premise that nitrate ions may be selection in ionization of neutral (or less acidic) clusters. To demonstrate the effect, they contrast the nitrate CIMS technique with acetate CIMS techniques, demonstrating that nitrate ion chemistry does not ionize all of the clusters generated in the flow reactor. The authors combine both experiment and simple models to describe the results.
The manuscript is well written, and should be accepted following the authors attention to the following minor comments:

Specific Comments:

I was surprised there was not a reference to the use of acetate ions for gasphase acid measurements (e.g., Veres et al. 2008 (Int. J. Mass Spectrom., doi:10.1016/j.ijms.2008.04.032, 2008)

I found the notations S160 / S125 (and similar) that are used throughout the figures to be confusing to the non-expert. I suggest defining these relationships in each figure caption. For example in Fig. 5, "Measured and modeled sulfuric acid-to-nitrate ion ratio (S160 / S125)" This helps keep the reader engaged and not flipping back to the definition in the manuscript. The same is true for S195 / S160 or S160 / S97.

Line 123-124: Are the reagent ion cluster distributions those observed in the mass spectrometer or those calculated to be in the source region. I would expect there to be a considerable difference between the reagent ions in the ion-molecule reaction region and those detected by the mass spectrometer following collisional dissociation. How might the reagent ion cluster size impact its ability to undergo proton transfer?

Line 126: What is the nominal cluster size used to calculate the assumed collision rate? Is there reason to suspect that the cluster size is different in nitrate and acetate mode?

**ACPD**

---

## Referee Comment (RC3) · Anonymous Referee #2 · 26 Aug 2016

This paper reports interesting experimental work on the behaviour of atmopsherically relevant cluster upon charging and subsequent travel though mass spectomteric instruments. The data has been analysed with ma modelling scheme, and result help to understand state of the art instruments in the field of atmospheric particle ormation as well as the formation process itself.

On top of the issues raised by the other two anonymous referees, I have only a few minor points that the authors could consider clarifyin:

Lines 32 &46: "...its clusters react with other trace compounds to produce stable electrically neutral ..." Is the idea here that the clusters are formed of sulphuric acid and water only, and the other compounds are added in the reactions mentioned? This could be specified, as now speaking of clusters existing before the reactions with trace

compounds sound a bit confusing

Line 55 Maybe add "that of" in the sentence "... proton affinity than THAT OF acetate..."

Line 236 Might be better to change "Following the neutral reactions . . . "-> "Following the neutral clustering reactions—"

Lines 343-348: "For all three diamines, we were unable to reproduce the observations with other combinations of Ââreactions and evaporation rates. The model only matched the observed trends when turning off the CI or formation of A2*diamine2.

Other explanations may exist to explain the differences between DMA and diamines observations (the most likely being semi-efficient [NITRATE?] CI of A2*diamine instead of zero nitrate CI of A2*diamine2), but additional thermochemical data (e.g., from more targeted experiments and computational chemistry) are needed to better inform the model. "

The explanation above feels slightly confusing as it seems that first it is stated that no other choice would lead to the observed trends, but then later another possibility is suggested. Could this be clarified? And could the word nitrate be added where I have inserted it in brackets in the above text?

---

## Author Comment (AC1)

We thank the reviewers for carefully reading and providing comments to improve our manuscript. Each point made by the reviewers is given as bold with our responses given in plain text. Specific lines in the manuscript are quoted with changes marked in red.

**Reviewer #1**

**The study by Jen et al. evaluates the capabilities of two different primary ions (nitrate and acetate) in ionizing clusters composed of sulfuric acid and dimethylamine (DMA), ethylene diamine (EDA), tetramethylethylene (TMEDA), or butanediamine/putrescine (PUT). Such clusters could in principle explain atmospheric new particle formation (NPF) since the produced neutral clusters have low evaporation rates.**

**The neutral clusters were formed in a flow reactor at ~300 K and at 30% relative humidity. A chemical ionization mass spectrometer (cluster CIMS) was used to detect the clusters after they reacted with nitrate or acetate primary ions used for the chemical ionization. Since the formed neutral clusters can contain equal numbers of acid and base molecules their reduced acidity could make them less susceptible towards ionization by nitrate in comparison to acetate primary ions.**

**Indeed, the presented results indicate that some clusters can very likely not be ionized efficiently by nitrate (e.g. the sulfuric acid dimer containing two diamines, or the trimer containing DMA or diamines).**

**In atmospheric studies nitrate chemical ionization is generally used for measuring sulfuric acid and its influence on NPF. If some atmospheric NPF is due to sulfuric acid and amines or diamines its importance could be significantly underestimated because the absence of sulfuric acid clusters would not necessarily indicate that sulfuric acid-amine NPF is not proceeding.**

**For this reason the manuscript by Jen et al. is a very important contribution and will help the interpretation of the mass spectra obtained in ambient and chamber CIMS measurements. The paper is very well-written and I have no serious comments. I therefore recommend its publication after addressing the points listed in the following:**

**Line 31: add a space before the bracket and also between the references (after the semicolon, please check the whole manuscript)**

Thank you for pointing this out. We have made every effort to correct these typos.

**Line 145: opening bracket is missing before Am.B2**

The bracket has been added.

**Line 171: "factor of 2 below", does this mean all concentrations are upper estimates?**

All the reported acetate concentrations represent an upper estimate. This is due to the uncertainty in the mass dependent sensitivity of for the smallest ions, including the acetate reagent ions. We have rephrased this sentence for clarity

"The systematic uncertainties of the acetate measurement are due to similar reasons as those for [$N_1$] and could lead to a factor of 2-3 times lower [$N_2$] than reported here."

**Figure 2: By comparing the panels a) and b) for the trimer it is not clear why the trimer in panel b) is >10 times higher for acetate than for nitrate. The trimer signals in panel a) are dominated by the cluster containing 3 acids and 1 base, however these signals seem to be quite similar for acetate and nitrate.**

We sincerely apologize for this graphical mistake. We have fixed Figure 2 and clarified the corresponding sentence in the text.

"For all bases, the measured [$N_3$] by acetate is 2 to 100 times higher than concentrations measured by nitrate CI."

[Figure]

**Figure 4: This figure seems to show signals for A− .Put, which is surprising given the fact that in the DMA system, A− .DMA evaporates very rapidly. It would be good to include some discussion about the presence of A− .Put.**

The reviewer points out a very interesting observation. We have wondered about this rather bizarre sighting of an aminated monomer. We do not see $A_1^-$•EDA or $A_1^-$•TMEDA either. Elm et al. (2016) very recently published free energies of sulfuric acid+diamine clusters. Though they did not model ions, the formation free energies of $A_1$•Put is lower at -15.4 kcal/mol than $A_1$•EDA at -11.1 kcal/mol; TMEDA was not modeled. $A_1$•DMA binding energy is -11.4 kcal/mol and is much closer to $A_1$•EDA. It is likely that the very strong binding energies of $A_1$•Put could mean that $A_1^-$•Put will survive until detection. More computational chemistry studies are needed to conclude this is the case.

Per the reviwer's suggestion, we have added a short paragraph discussing this.

"Unlike the other bases, Put was observed in the monomer using either nitrate or acetate CI (Figure 4). The presence of $A_1^-\bullet$Put indicates its binding energy must be higher than monomers containing the other bases. However, this ion still decomposes in roughly the $t_{CI}$=15 ms as it is ~0.1% of $[N_1]$. Elm et al. (2016) has shown that the binding energy of $A_1\bullet$EDA is -11.1 kcal/mol and $A_1\bullet$Put is -15.4 kcal/mol, with $A_1\bullet$DMA closely matching $A_1\bullet$EDA at -11.38 kcal/mol (Nadykto et al., 2014;Bork et al., 2014). The higher neutral binding energies of $A_1\bullet$Put may translate to stronger ion binding energies than the other aminated monomers, though more studies are needed to confirm this."

**Line 249 and SI equation S6: Equation S6 includes the difference between k21 (reaction between H2SO4 and HSO4 – ) and k1 (reaction between H2SO4 and NO3 – ) in the denominator of the equation. It seems that these values are identical (2x10-9 cm3 s-1), therefore this would lead to a zero division. Please clarify.**

The reviewer noticed an interesting point for equation S6. Two responses to this:

1) The forward rate constants used to invert Cluster CIMS signals to neutral cluster concentrations were not assumed to be all $2\text{x}10^{-9}$ $cm^3$ $s^{-1}$. For signal inversion, we used $k_1= 1.9\text{x}10^{-9}$ $cm^3$ $s^{-1}$ and $k_{21}=2\text{x}10^{-9}$ $cm^3$ $s^{-1}$. For the model, we used all forward rate constants as $2\text{x}10^{-9}$ $cm^3$ $s^{-1}$, but the model does not follow Equation S6 as it numerically solves all the cluster balance equations.

2) Equation S6 is a bit more complicated than $k_{21}-k_1$ in the denominator. This equation can be re-written as

$$\frac{\left[A_1^-\right]}{\left[NO_3^-\right]} = \frac{S_{160}}{S_{125}} = \frac{k_1}{k_{21}-k_1}\left(1-\exp\left(-\left(k_{21}-k_1\right)\left[N_1\right]t_{CI}\right)\right)$$

As $k_{21}-k_1$ becomes very small, we can do a Taylor series expansion on the exponential.

$$x = \left(-\left(k_{21} - k_1\right)\left[N_1\right]t_{CI}\right)$$

$$\frac{\left[A_1^-\right]}{\left[NO_3^-\right]} = \frac{k_1}{k_{21} - k_1}\left(1 - \left(1 + x + \frac{x^2}{2}\right)\right)$$

$$\frac{\left[A_1^-\right]}{\left[NO_3^-\right]} = \frac{-k_1}{k_{21} - k_1}\left(x + \frac{x^2}{2}\right)$$

$$\frac{\left[A_1^-\right]}{\left[NO_3^-\right]} = \frac{-k_1}{k_{21} - k_1}\left(\left(-\left(k_{21} - k_1\right)\left[N_1\right]t_{CI}\right) + \frac{\left(-\left(k_{21} - k_1\right)\left[N_1\right]t_{CI}\right)^2}{2}\right)$$

$$\frac{\left[A_1^-\right]}{\left[NO_3^-\right]} = \left(\left(k_1\left[N_1\right]t_{CI}\right) + \frac{k_1\left(k_{21} - k_1\right)\left(\left[N_1\right]t_{CI}\right)^2}{2}\right)$$

$$\lim_{k_{21} - k_1 \to 0} \frac{\left[A_1^-\right]}{\left[NO_3^-\right]} = \left(k_1\left[N_1\right]t_{CI}\right)$$

Therefore as $k_{21}-k_1$ becomes very small, the equation S6 becomes the equation typically used to convert signals to concentrations (Equation 1).

**Line 268 and line 246: two (slightly) different values for the ion-molecule reaction rates are given here. I would recommend to use the same value in the model and in equation 1.**

The modeled forward ion rate constants were taken to be $k_c=2\times10^{-9}$ cm$^3$ s$^{-1}$. As discussed in the SI, assuming all ion rate constants are equal does introduce error into the model, but this error is insignificant in comparison to uncertainties of the measurement and evaporation rate constants used in the model. We have tested the model with $k_1=1.9\times10^{-9}$ cm$^3$ s$^{-1}$ and the results were visually identical to those shown in Figure 5 and 6. Therefore, we prefer to use $k_c=2\times10^{-9}$ cm$^3$ s$^{-1}$ to keep the model notation simple.

**Line 293: please provide the value of the rate constant k21 here; this should be done for all rates by including their values in parentheses**

We have added the collision rate constant value.

"The rate constant, $k_{21}$, is the collisional rate constant of $2\times10^{-9}$ cm$^3$ s$^{-1}$."

**Line 329: "efficiently" instead of "inefficiently"?**

Fixed.

**Figure 7: The figure caption states a value of 4x109 cm-3 for the initial sulfuric acid concentration. However, in the legends different values for [A1] are given. Are these the concentrations after the 3 s reaction time? If so, please mention this in the figure caption.**

The reviewer is correct. We have added this clarification into the caption of the figures.

"Figure 6 Measured sulfuric acid dimer to monomer signal ratio ($S_{195}/S_{160}$) as a function of $t_{CI}$ for DMA (a), EDA (b), and TMEDA (c) measured by nitrate CI at $[A_1]_o \sim 4\times10^9$ cm$^{-3}$. The tables in panels a-c provide the measured $[A_1]$ at that $[B]$ after the 3 s acid-base reaction time. Observations were fitted according to Equation 2 with the y-intercept shown by the dashed line. Panels d-f present modelled results for each base.

"Figure 7 Measured sulfuric acid dimer to monomer signal ratios ($S_{195}/S_{160}$ for nitrate or $S_{97}$ for acetate) as a function of CI reaction time using nitrate (a) and acetate CI (c). In both cases, $[A_1]_o$ was held constant at $4\times10^9$ cm$^{-3}$. Panel (b) shows the modeled results for Put. The table inside panel (a) and (c) provide the measured $[A_1]$ after the 3 s acid-base reaction time."

**Line 374: Do these evaporation rates refer to the evaporation of DMA or A?**

These are DMA evaporation rates.

"CI of $N_3$ leads to ions such as (i) $A_3^-\cdot DMA_3$ which evaporate at a rate of $10^4$ s$^{-1}$ into $A_3^-\cdot DMA_2$ and (ii) $A_3^-\cdot DMA_2$ and $A_3^-\cdot DMA$ which have predicted DMA evaporation rates of $\sim 10^{-1}$ and $10^{-2}$ s$^{-1}$"

**Line 425: better use "sources of dimer ions" than "dimer ion sources"**

Agreed—much better phrasing.

"These reactions have little effect on the modeled dimer results since they introduce minor sources of dimer ions."

**Table 3: for the neutral pathway it seems the reaction A3B2 + B is missing**

This pathway is indeed included in our model. We added this missing reaction in Table 3 and Table S2.

$$A_2 \bullet B + A_1 \xrightarrow{k} A_3 \bullet B$$

$$A_3 \bullet B + B \xrightarrow{k} A_3 \bullet B_2$$

$$A_3 \bullet B_2 + B \xrightarrow{k} A_3 \bullet B_3$$

$$A_2 \bullet B_2 + A_1 \xrightarrow{k} A_3 \bullet B_2$$

$$A_2 \bullet B + A_1 \bullet B \xrightarrow{k} A_3 \bullet B_2$$

$$A_2 \bullet B_2 + A_1 \bullet B \xrightarrow{k} A_3 \bullet B_3$$

**Line 466: do you mean "low" instead of "high"?**

We do mean high. Since the [DMA] is much lower than $[A_1]$, we do not anticipate seeing $[A_5\cdot DMA_5]$ as high as $10^7$ cm$^{-3}$. We expect to see pentamers with less than 5 DMA, closer to 3 or even 4.

**References: please use same style for all references, i.e. remove hyperlinks and add page numbers for all references, etc.**

We have standardized the references to follow ACP guidelines. Unfortunately, Endnote does not track changes so it is difficult to show the changes in the manuscript.

**SI, Line 39: "cm-3" instead of "cm3 ", also better to use "s-1" instead of "Hz"**

Thanks for catching that typo. However, we would prefer to use Hz cm$^{-3}$ as Hz better implies signal of the MS, and this would keep our measurements consistent with the sensitivity curves Zhao et al. (2010) reported.

**SI, Equation S6: see comment above**

See response above.

**Table S1: In a previous paper by the same authors (Jen et al., 2016, GRL) it was concluded that diamines are a more potent source of new particles in comparison with DMA. However, from the evaporation rates listed in Table 1 this conclusion does not seem to be supported due to the rather high evaporation rates E1 for the diamines (50 times higher than for DMA) and the crucial role of A1B1 in terms of cluster formation. How can this discrepancy be explained and how does it affect the conclusions from the present paper?**

The evaporation rates listed in Table S1 are not the true evaporation rates for two reasons:

1) These are just evaporation rates used in our modeling. The model is not perfect as it assumes certain pathways for cluster formation to reduce the number of fitted parameters. A more comprehensive model (like ACDC) is need to capture all possible pathways.

2) Even with our assumed model, these evaporation rates imply that the cluster lifetimes are on the order of the 3 s acid-base neutral reaction time. Our experiment does not have the time sensitivity to distinguish between clusters with lifetimes differing by a few seconds.

Regardless of the very large uncertainty in these evaporation rates, consideration must also be given to the evaporation rates for the larger clusters. The formation of $A_1 \cdot B_1$ is very crucial but assuming the evaporation of $A_1 \cdot B_1$ is the bottleneck to nucleation ($E_{2,3}$=0 s$^{-1}$) will fail to distinguish the differences between DMA and the diamines. Two DMA molecules are needed to form a cluster without appreciable evaporation rates whereas just one diamine molecule is needed. This is key conclusion of Jen et al. (2016) and is further confirmed by computational results from Elm et al. (2016). There is no discrepancy between Jen et al. (2016) and this paper. In fact, they both tell the same story. The formation of $A_1 \cdot$Diamine is likely the bottleneck to nucleation for these clusters whereas there are multiple bottlenecks to nucleation for sulfuric+DMA clusters.

We have mentioned both the large uncertainty of the evaporation rates and how this study relates to Jen et al. (2016) in the manuscript and SI. Though the reviewer brings up a subtle and very important point, we have decided to not add any more lines to the paper.

**Reviewer #2**

This paper reports interesting experimental work on the behaviour of atmospherically relevant cluster upon charging and subsequent travel though mass spectomteric instruments. The data has been analysed with ma modelling scheme, and result help to understand state of the art instruments in the field of atmospheric particle formation as well as the formation process itself.

On top of the issues raised by the other two anonymous referees, I have only a few minor points that the authors could consider clarifying:

Lines 32 &46: "...its clusters react with other trace compounds to produce stable electrically neutral ..." Is the idea here that the clusters are formed of sulphuric acid and water only, and the other compounds are added in the reactions mentioned? This could be specified, as now speaking of clusters existing before the reactions with trace compounds sound a bit confusing

The intent of that sentence is to say that sulfuric acid and cluster containing sulfuric acid (which could also contain water, ammonia, etc.) react with trace gases to form stable, electrically neutral clusters. We have clarified our meaning in these two lines (references removed to make it easier to read).

"In the atmospheric boundary layer, sulfuric acid often participates in nucleation by reacting with other trace compounds to produce stable, electrically neutral molecular clusters"

"The first process, neutral cluster formation, follows a sequence of acid-base reactions whereby sulfuric acid vapor and its subsequent clusters react with basic molecules to produce clusters that are more stable than aqueous sulfuric acid clusters."

Line 55 Maybe add "that of" in the sentence "... proton affinity than THAT OF acetate..."

Agreed.

Line 236 Might be better to change "Following the neutral reactions . . . "-> "Following the neutral clustering reactions—"

Agreed.

Lines 343-348: "For all three diamines, we were unable to reproduce the observations with other combinations of reactions and evaporation rates. The model only matched ˘ the observed trends when turning off the CI or formation of A2*diamine2.

Other explanations may exist to explain the differences between DMA and diamines observations (the most likely being semi-efficient [NITRATE?] CI of A2*diamine instead of zero nitrate CI of A2*diamine2), but additional thermochemical data (e.g., from more targeted experiments and computational chemistry) are needed to better inform the model. "

**The explanation above feels slightly confusing as it seems that first it is stated that no other choice would lead to the observed trends, but then later another possibility is suggested. Could this be clarified? And could the word nitrate be added where I have inserted it in brackets in the above text?**

Yes, it is a bit confusing. We have reworded this section.

"For all three diamines, we were unable to reproduce the observations with other combinations of reactions and evaporation rates. The model only matched the observed trends by turning off the CI or formation of $A_2\bullet diamine_2$.

However, several of the modeled reactions are simplified versions of multi-step reactions. For example, preventing the formation of $A_2\bullet TMEDA_2$ could also mean $A_2\bullet TMEDA_2$ forms at the collision rate but instantly decomposes into $A_2\bullet TMEDA$. Furthermore, differences between DMA and diamine observations could instead be explained by semi-efficient nitrate CI of $A_2\bullet diamine$ because the existence of high $[A_2\bullet diamine_2]$ is unlikely due to its high basicity. Preventing $A_2\bullet diamine_2$ from forming and semi-efficient CI of $A_2\bullet diamine$ could lead to identical results as shown in the model for EDA and TMEDA. Additional thermochemical data (e.g., from more targeted experiments and computational chemistry) are needed to better inform the model. Regardless, our observations and modeling show that dimer's neutral formation pathways and/or the nitrate CI differs between the DMA and diamine systems."

**Reviewer #3**

**Review of Jen et al., Chemical ionization of clusters formed from sulfuric acid and dimethylamine or diamines**

**Summary and General Comments: The authors present a series of experiments designed to assess the utility of nitrate ion CIMS techniques for the detection of H2SO4- base clusters that lead to the formation of new particles in the atmosphere. Nitrate ion CIMS has been used for detection of low vapor pressure trace gases previously, and is well known to be a highly sensitive, but very specific reagent ion. The authors extend this logic to the detection of clusters, under the premise that nitrate ions may be selection in ionization of neutral (or less acidic) clusters. To demonstrate the effect, they contrast the nitrate CIMS technique with acetate CIMS techniques, demonstrating that nitrate ion chemistry does not ionize all of the clusters generated in the flow reactor. The authors combine both experiment and simple models to describe the results.**

**The manuscript is well written, and should be accepted following the authors attention to the following minor comments:**

**Specific Comments: I was surprised there was not a reference to the use of acetate ions for gasphase acid measurements (e.g., Veres et al. 2008 (Int. J. Mass Spectrom., doi:10.1016/j.ijms.2008.04.032, 2008)**

Veres et al. is indeed the seminal paper presenting the use of acetate to chemically ionize gas phase acids. We should note that Veres et al. differs from this study in that we are chemically

ionizing with acetate at atmospheric pressure. We have added the reference to illustrate the differences of proton affinities for acetate compared to nitrate.

"Acetate CI has been used previously to detected organic acids less acidic than sulfuric acid in the atmosphere, providing evidence that its higher proton affinity could chemically ionize more basic clusters (Veres et al., 2008). Subsequently, Jen et al. (2015) showed that CI with $(HNO_3)_{1-2} \cdot NO_3^-$ leads to significantly lower neutral concentrations of clusters with 3 or more sulfuric acid molecules and varying numbers of DMA molecules compared to results using acetate reagent ions."

**I found the notations S160 / S125 (and similar) that are used throughout the figures to be confusing to the non-expert. I suggest defining these relationships in each figure caption. For example in Fig. 5, "Measured and modeled sulfuric acid-to-nitrate ion ratio (S160 / S125)" This helps keep the reader engaged and not flipping back to the definition in the manuscript. The same is true for S195 / S160 or S160 / S97.**

We agree that this notation can be confusing to those not familiar with the masses of the clusters (which is the vast majority of the readers). We have taken the advice of the reviewer and added those clarifying remarks.

"Figure 5 Measured (a,b) and modeled (c, d) sulfuric acid monomer to nitrate signal ratio $(S_{160}/S_{125})$ as a function of CI reaction time for DMA (a, c) and EDA (b, d). The measurements were conducted with nitrate as the reagent ion and at $[A_1]_o \sim 4 \times 10^9$ cm$^{-3}$. Each color represents a different [B] with the linear regressions for the measurements given in colored text."

"Figure 6 Measured sulfuric acid dimer to monomer signal ratio $(S_{195}/S_{160})$ as a function of $t_{CI}$ for DMA (a), EDA (b), and TMEDA (c) measured by nitrate CI at $[A_1]_o \sim 4 \times 10^9$ cm$^{-3}$. The tables in panels a-c provide the measured $[A_1]$ at that [B] after the 3 s acid-base reaction time. Observations were fitted according to Equation 2 with the y-intercept shown by the dashed line. Panels d-f present modelled results for each base. "

"Figure 7 Measured sulfuric acid dimer to monomer signal ratios $(S_{195}/S_{160}$ for nitrate or $S_{97}$ for acetate) as a function of CI reaction time using nitrate (a) and acetate CI (c). In both cases, $[A_1]_o$ was held constant at $4 \times 10^9$ cm$^{-3}$. Panel (b) shows the modeled results for Put. The table inside panel (a) and (c) provide the measured $[A_1]$ after the 3 s acid-base reaction time."

"Figure 8 Measured bare sulfuric acid trimer to monomer signal ratio $(S_{293}/S_{160})$ as a function of $t_{CI}$ for DMA (a), EDA (b), and TMEDA (c) detected by nitrate CI at $[A_1]_o = 4 \times 10^9$ cm$^{-3}$."

"Figure 9 Nitrate measured signal ratio between $A_3 \cdot B$ and sulfuric acid monomer $(S_{A3 \cdot B}/S_{160})$ as a function of $t_{CI}$ for DMA (a), EDA (b), and TMEDA (c) at $[A_1]_o = 4 \times 10^9$ cm$^{-3}$."

"Figure 10 Nitrate measured signal ratio between $A_4 \cdot B$ and sulfuric acid monomer $(S_{A4.diamine}/S_{160})$ as a function of CI reaction time for EDA (a), Put (b), and TMEDA (c)."

**Line 123-124: Are the reagent ion cluster distributions those observed in the mass spectrometer or those calculated to be in the source region. I would expect there to be a considerable difference between the reagent ions in the ion-molecule reaction region and**

**those detected by the mass spectrometer following collisional dissociation. How might the reagent ion cluster size impact its ability to undergo proton transfer?**

The reagent ion distribution is measured by the Cluster CIMS. We believe the measured distribution does reflect the makeup in the ion-molecule reaction region as we tuned the Cluster CIMS to minimize cluster fragmentation. Measurements on nitrate cluster binding enthalpies supports our observations for the nitrate ion distribution, with dimer ion being the most strongly bonded and thus the highest in signal/concentration (Lovejoy and Bianco, 2000). In addition, the Cluster CIMS does not have a specific collisional dissociation chamber (CDC) like most CIMS instruments; instead it only has conical octopoles to focus the ion clusters prior to the quadrupole. With that being said, we are still uncertain on the exact composition of the reagent ion clusters, i.e. what else is attached to the nitrate clusters. Tanner and Eisele (1995) showed RH does not affect nitrate dimer chemical ionization of sulfuric acid. At high base concentrations and low sulfuric acid concentrations, base molecules cluster with the reagent ions, as shown in Simon et al. (2016). It still is not known how base ligands on the reagent ions affect its chemical ionization abilities.

We have explicitly written out our assumption about treating the nitrate dimer and trimer (and all the acetate ions) as the essentially the same ion during CI. We do take into account mass dependent sensitivity which is explained in the SI.

"The measured reagent ions for nitrate CI was $(HNO_3)_{1-2} \cdot NO_3^-$, and the reagent ions for acetate CI were $H_2O \cdot CH_3CO_2^-$, $CH_3CO_2H \cdot CH_3CO_2^-$, and $CH_3CO_2^-$ (in order of abundance). The nitrate dimer and trimer are assumed to chemically ionize at equal rate constants, and the three acetate ions are assumed to chemically ionize in identical manners."

**Line 126: What is the nominal cluster size used to calculate the assumed collision rate? Is there reason to suspect that the cluster size is different in nitrate and acetate mode?**

The rate constant for the chemical ionization of sulfuric acid and nitrate dimer has been measured and is $1.9 \times 10^{-9}$ cm$^3$ s$^{-1}$. The measured trimer rate constant is a bit slower at $1.7 \times 10^{-9}$ cm$^3$ s$^{-1}$ but this number is uncertain due to ion decomposition at T>273 K (Viggiano et al., 1997). To our knowledge, the dipole moments of acetate and its clusters has not been measured. Though acetate dimer and monomer are about the same mass as nitrate dimer and monomer, respectively, the dipole moments will have a larger influence on the collision rate than mass ($m^{-1/2}$ vs. $\mu$) (Su, 1973). Thus, we have assumed acetate and nitrate have similar collision rate constants with sulfuric acid. Such small differences between collision rate constants would not explain the very large differences between the cluster concentrations detected by nitrate and acetate.

**References cited in this response:**

Elm, J. and Jen, C. N.: Strong Hydrogen Bonded Molecular Interactions between Atmospheric Diamines and Sulfuric Acid, The Journal of Physical Chemistry A, 120(20), 3693, doi:10.1021/acs.jpca.6b03192, 2016.

Lovejoy, E. R. and Bianco, R.: Temperature Dependence of Cluster Ion Decomposition in a Quadrupole Ion Trap †, The Journal of Physical Chemistry A, 104(45), 10280, doi:10.1021/jp001216q, 2000.

Simon, M., Heinritzi, M., Herzog, S., Leiminger, M., Bianchi, F., Praplan, A., Dommen, J., Curtius, J. and Kurten, A.: Detection of dimethylamine in the low pptv range using nitrate chemical ionization atmospheric pressure interface time-of-flight (CI-APi-TOF) mass spectrometry, Atmospheric Measurement Techniques, 9(5), 2135, doi:10.5194/amt-9-2135-2016, 2016.

Su, T.: Theory of ion- polar molecule collisions. Comparison with experimental charge transfer reactions of rare gas ions to geometric isomers of difluorobenzene and dichloroethylene, The Journal of Chemical Physics, 58(7), 3027–3037, doi:doi:http://dx.doi.org/10.1063/1.1679615, 1973.

Tanner, D. J. and Eisele, F. L.: Present OH measurement limits and associated uncertainties, Journal of Geophysical Research, 100(D2), 2883, doi:10.1029/94jd02609, 1995.

Viggiano, A. A., Seeley, J. V. and Mundis, P. L.: Rate Constants for the Reactions of $XO_3^-$ $(H_2O)_n$ (X = C, HC, and N) and $NO_3^-$ $(HNO_3)_n$ with $H_2SO_4$ :Implications for Atmospheric Detection of $H_2SO_4$, The Journal of …, 1997.

Zhao, J., Eisele, F. L., Titcombe, M., Kuang, C. and McMurry, P. H.: Chemical ionization mass spectrometric measurements of atmospheric neutral clusters using the cluster-CIMS, Journal of Geophysical Research, 115(D8), doi:10.1029/2009jd012606, 2010.